# TRUTH TABLE DEEP CONVOLUTIONAL NEURAL NETWORK, A NEW SAT-ENCODABLE ARCHITECTURE - APPLICATION TO COMPLETE ROBUSTNESS

## ABSTRACT

With the expanding role of neural networks, the need for formal verification of their behavior, interpretability and human post-processing has become critical in many applications. In 2018, it has been shown that Binary Neural Networks (BNNs) have an equivalent representation in boolean logic and can be formally analyzed using logical reasoning tools such as SAT or MaxSAT solvers. This formulation is powerful as it allows us to address a vast range of questions: existential, probabilistic, explanation generation, etc. However, to date, only BNNs can be transformed into a SAT formula and their strong binary constraints limit their natural accuracy. Moreover, the corresponding SAT conversion method intrinsically leads to formulas with a large number of variables and clauses, impeding interpretability as well as formal verification scalability. In this work, we introduce Truth Table Deep Convolutional Neural Networks (TT-DCNNs), a new family of SAT-encodable models featuring real-valued weights and real intermediate values as well as a highly interpretable conversion method. The TT-DCNN architecture enables for the first time all the logical classification rules to be extracted from a performant neural network which can be then easily interpreted by anyone familiar with the domain. Therefore, this allows integrating human knowledge in post-processing as well as enumerating all possible inputs/outputs prior to deployment in production. We believe our new architecture paves the way between eXplainability AI (XAI) and formal verification. First, we experimentally show that TT-DCNNs offer a better tradeoff between natural accuracy and formal verification than BNNs. Then, in the robustness verification setting, we demonstrate that TT-DCNNs outperform the verifiable accuracy of BNNs with a comparable computation time. Finally, we also drastically decrease the number of clauses and variables, enabling the usage of general SAT solvers and exact model counting solvers. Our developed real-valued network has general applications and we believe that its demonstrated robustness constitutes a suitable response to the rising demand for functional formal verification.

## 1 INTRODUCTION

Deep Neural Network (DNN) systems offer exceptional performance in a variety of difficult domains (Goodfellow et al., 2016) and today these results far outstrip our ability to secure and analyze those DNNs. As DNNs are becoming widely integrated in a variety of applications, several concerns have emerged: lack of robustness emphasized by a lack of explainability, difficulty of integrating human knowledge in post-processing and impossibility to formally verify their behavior due to their large complexity. Under these circumstances and especially when these systems are deployed in applications where safety and security are issues, the formal verification of DNN systems and the field of eXplainable AI (XAI) are under intense research efforts. For example, Tesla has recently filed a patent on the DNNs' portability on a platform

incorporating a component dedicated to formal verification (Driscoll, 2020). Also, the European Union's general data protection regulation includes a provision on the explainability of AIs (Regulation, 2016).

DNNs formal verification methods are mainly based either on Satisfiability Modulo Theory (SMT) (Katz et al., 2017) or Mixed-Integer Programming (MIP) (Xiao et al., 2018) which are not yet scalable to real-valued DNNs. Some recent publications (Jia & Rinard (2020);Narodytska et al. (2019b); Narodytska et al. (2018)) approach the problem of complete verification from the well-known boolean SATisfiability (SAT) (Biere et al., 2009) point of view where Binary Neural Networks (BNNs, Hubara et al. (2016)) are first converted into SAT formulas and then formally verified using SAT or MaxSAT solvers. This pipeline is computationally efficient (Jia & Rinard, 2020), enables security verifications (Baluta et al., 2019) and more generally can answer a vast range of question such as how many adversarial attacks exist for a given DNN, image and noise level (Narodytska et al., 2019a). Besides, this approach is faster than SMT or MIP robustness verification method (Jia & Rinard, 2020). However, to date, only BNNs can be transformed into a SAT formula and their strong binary constraints limit their natural accuracy. Moreover, the corresponding SAT conversion method intrinsically leads to formulas with a large number of variables and clauses, impeding interpretability as well as formal verification scalability. Finally, only few studies (Ignatiev et al. (2019a) ; Ignatiev et al. (2019b)) investigated the relationship between formal DNNs' verification and XAI.

**Our contributions.** In this work, we offer three main contributions. **(1)** First, we define a new family of real-valued Deep Convolutional Neural Networks (DCNN) that can be encoded into SAT formulas: Truth Table Deep Convolutional Neural Network (TT-DCNN). Our TT-DCNN leverages its model formulation in the form of a truth table to allow weights and certain intermediate values to be real. To the best of our knowledge, this is the first method to encode a real-valued DCNN into SAT. For the first time, we can extract all the logic classification rules from a subfamily of DCNNs which allows a bridge between XAI and formal verification, while achieving sufficient natural accuracy for practical use. Indeed, the nature of the SAT conversion between TT-DCNN and BNN is intrinsically different: our method relies upon giving one SAT expression per 2D-CNN filter instead of one SAT expression per neuron. This global interpretability method is in sharp contrast with the previous limited BNNs and local DNNs explainability. **(2)** TT-DCNNs offer two main valuable conversion properties over BNNs. **(2-a: Post-tuning)** The first one allows us to integrate human knowledge in the post-processing: we can now interpret the model inference with simple concepts, which enables to manually modify the 2D-CNN filter activation towards a desired goal. For example, we decided to focus on reducing overfitting and, to this end, we characterize TT-DCNN logic rules resulting from overfitting and propose a filtering approach, which increases the verifiable accuracy without decreasing the natural accuracy (cf. Appendix A.1). **(2-b: Tractability)** The second property is the possibility to compute all possible model inputs/outputs prior to deployment in production. In an adversarial setting, we can assess whether the input noise will propagate to the output. We can therefore disregard filters with no impact on the output. This leads to a lower number of clauses and variables in the SAT formulas compared to BNNs which allows using generic SAT solvers and exact model counting solvers. **(3)** We apply our model to complete robustness verification (cf. Appendix A.1). TT-DCNNs offer a good tradeoff between the state-of-the-art of BNN/SAT method (Jia & Rinard, 2020) and of real-valued DNN/MILP complete robustness verification methods (Xiao et al. (2018); Tjeng et al. (2017)). This is expected as our network is both real-weighted and SAT-convertible. Our TT-DCNN model improves the verifiable accuracy by more than 2.5% for high noise MNIST and by 0.5% for the high noise of CIFAR10 when compared to BNN/SAT method while decreasing the verification time by a factor of 9 for MNIST and 150 for CIFAR10 high noise when compared to DNN/MILP methods. Finally, our SAT formulas are 5 and 9 times more compact in term of number of clauses for high noise MNIST and CIFAR10 respectively compared to the BNN/SAT method.

**Outline.** Section 2 introduces the notations and the related work. Section 3 presents our new TT-DCNN model and its two main properties. Section 4 details the complete robustness verification set-up and reports the evaluation results. Finally, we conclude this work in Section 5.

## 2 BACKGROUND & RELATED WORK

**Boolean SATisfiability (SAT).** The boolean SATisfiability problem (SAT) (Biere et al., 2009) is the problem of deciding whether there exists a variable assignment to satisfy a given boolean expression $\Phi$. We can consider a boolean expression in a Conjunctive Normal Form (CNF) or in a Disjunctive Normal Form (DNF). They are both defined over a set of boolean variables $(x_1, \cdots, x_n)$. A literal $l_i$ is defined as a variable $x_i$ or its complement $\overline{x_i}$. A CNF is a conjunction of a set of clauses: $\Phi = (c_1 \wedge \cdots \wedge c_m)$, where each clause $c_j$ is a disjunction of some literals $c_j = l_{j1} \vee \cdots \vee l_{jr}$. A DNF is a disjunction of a set of clauses: $\Phi = (c_1 \vee \cdots \vee c_m)$, where each clause $c_j$ is a conjunction of some literals $c_j = l_{j1} \wedge \cdots \wedge l_{jr}$. A pseudo-boolean constraint is a constraint of the form: $\sum_{p=1}^{N} a_p l_p \circ b$, where $a_p \in \mathbb{Z}$, $b \in \mathbb{Z}$ and $\circ \in \{\leq, =, \geq\}$, which can be mapped to a SAT formula (Roussel & Manquinho, 2009). However, the output SAT formula contains a tremendous number of clauses and literals compared to the number of variables in the pseudo-boolean constraint making it very hard to understand (cf. example in Appendix A.2). A boolean function has the form $\{0, 1\}^n \rightarrow \{0, 1\}$ and its corresponding truth table gives outputs for all possible inputs combinations.

**Two-dimensional Convolutional Neural Networks (2D-CNNs).** We consider the 2D-CNN as a function $\Phi_f$, which, for a given filter $f$, takes $n = k^2 c$ inputs at position $(i, j)$ with $k$ the kernel size and $c$ the number of input channels. The outputs can be written $y_f^{(i,j)} = \Phi_f(x_1^{(i,j)}, \cdots, x_n^{(i,j)})$. Note that in the binary case, a truth table between inputs and outputs for $2^n$ entries can be easily set up (if $n$ is not too large). If we now consider a multi-layers network with $s$ convolution layers : a similar truth table can be constructed, except now the kernel size $k$ needs to be replaced by a *patch function $P(s)$* (also referred to as the size of a receptive field in the literature). We have $P(1) = k$ and $P(s + 1) = P(s)$ if and only if the kernel size of the layer is 1. We denote by $P(s, i, j)$, the receptive field after the $s^{th}$ layer at position $(i, j)$. We denote the vector obtained after the flatten operation and before the final classifier layer as the vector of features $V$. If there are a total of $L$ layers in the DNN, each element of $V$ is a non-linear function over a patch $(P(L), P(L))$ on the input.

**SAT encoding of neural networks.** The sole published method converting a DNN into a SAT formula is limited to BNNs (Narodytska et al. (2018); Cheng et al. (2018)) and involves recomposing a block formed of a 2D-CNN layer, a batch normalization layer and a step function into an inequality in order to apply the pseudo-boolean constraint (Roussel & Manquinho, 2009). This approach has been further refined using a different training method and a specific SAT solver resulting in a significantly reduced verification resolution time (Jia & Rinard, 2020). Although the proposed inequality rewriting is elegant, the corresponding SAT formula contains a tremendous number of clauses and literals compared to the number of variables in the pseudo-boolean constraint (cf. example Appendix A.2). This prevents both the interpretability and the tractability of those SAT/BNNs formulas. Hence, our goal is to find a real-valued DCNN with good performance and coincidentally convertible to SAT with fully interpretive inference.

**Interpretability by global rule extraction.** Machine learning interpretability analysis fall into four main categories: either local (input dependant) or global methods, either exact or non-exact methods. The most famous techniques for local non-exact interpretability are LIME and the ANCHOR explainers (Ribeiro et al. (2016); Ribeiro et al. (2018)). PI-explanations (Shih et al., 2018) and SHAP (Lundberg & Lee, 2017) are also popular techniques for local exact method and global non-exact method respectively. The only scalable method for global exact interpretability was proposed in (Granmo et al., 2019). Our work aims to extend the studies of the latter strategy with the use of truth tables in order to obtain the equivalent conjectures. Using truth tables in machine learning has been documented for hardware optimisation (Soga & Nakahara (2020), Wang et al. (2019)) and recently in an attempt for global non-exact interpretation of BNNs (Burkhardt et al., 2021). Our work is pioneer in the use of truth table to create a new architecture enabling the extraction of all the global exact logic rules as well as the increase of model robustness for formal verification.

## 3 TRUTH TABLE DEEP CONVOLUTION NEURAL NETWORK (TT-DCNN)

In an attempt to address the currently identified drawbacks of the low interpretability and the high encoding complexity of the SAT formulas of the BNN's transformation process, we designed a new DNN architecture. Our TT-DCNN model is developed first as a DCNN with real-valued weights and some intermediate real values encodable into SAT. Secondly, TT-DCNN is interpretable: each feature can be understood as an indicator function. In fact, the feature associated to the filter $f$ is equal to 1 if there exists a mask $M$ in the set of masks $\mathcal{S}_f$ that matches the input patch, 0 otherwise (we will define later what are a mask, a set of masks and the matching operator). We emphasize that our formulation is a new paradigm: there is no longer any need for the DNNs in production. Since the knowledge of the TT-DCNN is reduced to the set $\mathcal{S}_f$ for all filters, we no longer need the model after the training. To the best of our knowledge, this is the first time that the exact knowledge of a real-valued DCNN is formally extracted. Finally, we will show that this formulation gives the user two major degrees of freedom: post-tuning and tractability.

Briefly, our model considers an image with three channels and floating-point inputs, while being composed of only one block of 2D-CNN as defined in Section 3.2. In order to convert the image into binary inputs, we incorporate a preprocessing layer before this block. Finally, we include a final linear layer after the 2D-CNN block to perform classification. For clarity sake, we will present our new TT-DCNN model in a stepwise fashion with increasing complexity using illustrative examples and a companion video [1]. We will initially analyse a 2D-CNN layer with a single filter (Section 3.1). Next, we will define a block of our TT-DCNN architecture (Section 3.2). Finally, Sections 3.3 and 3.4 present the whole architecture of the TT-DCNN and its main properties, respectively.

### 3.1 SAT ENCODING OF A ONE-LAYER 2D-CNN

For this first building block, we consider as input a binary image with one channel and as a model a trained DCNN with only one layer. We first start to encode one filter.

**One filter.** The main idea is to fix the number of possible outputs of the 2D-CNN by fixing the number of its possible inputs, which will allow to test all possible combinations. Following the notations introduced in Section 2, we have as output $y_{binary,f}^{(i,j)}$, for the filter $f$:

$$y_{binary,f}^{(i,j)} = Bin(y_f^{(i,j)}) = Bin(\Phi_f(x_1^{(i,j)}, \cdots, x_n^{(i,j)})) \tag{1}$$

with $Bin$ being the Heaviside step function, defined as $Bin(x) = \frac{1}{2} + \frac{sgn(x)}{2}$. If $x_j \in \{0, 1\}$, we can establish the truth table of the 2D-CNN's filter $f$ by trying all the possible inputs, for a total of $2^n$ operations. In this paper, we will limit ourselves to $n \leq 9$. Hence, in $2^9 = 512$ operations, we can trivially generate our truth table. Then, we can convert the truth table into a simplified SAT formula and by doing so we can rewrite Equation 1 as:

$$y_{binary,f}^{(i,j)} = \mathsf{SAT}_f^{\mathsf{DNF}}(x_1^{(i,j)}, \cdots, x_n^{(i,j)}) = \mathsf{SAT}_f^{\mathsf{CNF}}(x_1^{(i,j)}, \cdots, x_n^{(i,j)})$$

with $\mathsf{SAT}_f^{\mathsf{DNF}}$ (resp. $\mathsf{SAT}_f^{\mathsf{CNF}}$) being the formal expression of the filter in the DNF form (resp. CNF form). It is noteworthy that unlike previous works, our approach is not limited to binary weights but allows for arbitrary weights within the 2D-CNN.

**Example.** We consider a 2D-CNN with one filter and a kernel size of 2, with the weights: $W_1 = \begin{bmatrix} 10 & -1 \\ 3 & -5 \end{bmatrix}$. As $c = 1$, we have $X = \begin{bmatrix} x_0 & x_1 \\ x_2 & x_3 \end{bmatrix}$ and the sixteen possible entries are: $\begin{bmatrix} 0 & 0 \\ 0 & 0 \end{bmatrix}$,

---

[1]Link video: https://youtu.be/loGlpVcy0AI

$\begin{bmatrix} 0 & 0 \\ 0 & 1 \end{bmatrix}, \begin{bmatrix} 0 & 0 \\ 1 & 0 \end{bmatrix}, \begin{bmatrix} 0 & 0 \\ 1 & 1 \end{bmatrix}, \cdots, \begin{bmatrix} 1 & 1 \\ 0 & 1 \end{bmatrix}, \begin{bmatrix} 1 & 1 \\ 1 & 1 \end{bmatrix}$. For each input, we calculate the corresponding output: $y = [0, -5, 3, -2, -1, -5, 3, -2, 10, 5, 13, 8, 9, 4, 12, 7]$. After binarization with the Heaviside step function, we have $y_{binary} = [0, 0, 1, 0, 0, 0, 1, 0, 1, 1, 1, 1, 1, 1, 1, 1]$. Therefore, after simplification, we have $\mathsf{SAT}_1^{\mathsf{DNF}} = (x_2 \wedge \overline{x_3}) \vee x_0$ and $\mathsf{SAT}_1^{\mathsf{CNF}} = (x_2 \vee x_0) \wedge (\overline{x_3} \vee x_0)$.

**Multiple filters.** For multiple filters, the above described method is simply repeated for each individual filter, thus yielding one expression per filter.

**Multiple channels.** As convolutional networks generally take several channels as input (3 for RGB images or more for intermediate 2D-CNNs) the number of input variables can therefore greatly increase. A 2D-CNN that takes 32 input channels with a kernel size of 2 yields an input of size 128, well above our limit set at $n = 9$. To overcome this, we gather the channels by groups (Dumoulin & Visin, 2016). Grouped convolutions divide the input channels into $g$ groups, then apply separate convolutions within each group; this effectively decreases the number of inputs to each individual filter by a factor of $g$. We have in that case $n = k^2 c/g$. Thus, in the above example, by using 16 groups, the number of inputs to our truth tables becomes $\frac{32}{16} \times 2^2 = 8$.

**Limits.** At this point of the model development and despite the real-valued weights, we observe sub-optimal performances due to the group parameters. When we add an extension layer, as detailed in the next subsection, to increase the learning capacity of the DCNN without augmenting the size of the patches seen by the DCNN (Sandler et al., 2018), we experience an improvement of about 5% of natural accuracy. Please refer to Table 8 of Appendix A.9.2 for detailed results.

## 3.2 SAT ENCODING OF A TWO-LAYERS 2D-CNN: ONE BLOCK OF TT-DCNN

**Amplification layer.** In the previous subsection, we pointed out that only the 2D-CNN input size matters when establishing the 2D-CNN SAT expression. Therefore, we can add a second layer as long as we do not increase the patch size. This can be simply achieved by adding a layer with kernel size 1. Note that the intermediate values from the first layer do not need to be binary anymore. From a practical standpoint, using eight filters in layer one and one filter in layer two, drastically improves the learning capacity of the TT-DCNN and therefore the natural accuracy as well. This observation is consistent with other published studies (Sandler et al., 2018). A block is therefore composed of two 2D-CNN layers with a so-called amplification parameter which corresponds to the ratio between the number of filters of the first layer and the number of filters of the second layer (value often set at 8). Each layer is followed by a batch normalisation and a non-linear activation function: ReLU for the first layer and Heaviside step function for the second one.

## 3.3 THE TT-DCNN ARCHITECTURE

Having defined one block of 2D-CNN in the previous section, we may now examine how this block is integrated into the TT-DCNN. The overall architecture is presented in Appendix A.3.1

**Preprocessing layer.** DCNNs inputs are usually floating points. However, encoding floating points in SAT typically implies high complexity. In order to simplify the verification process and improve the network robustness, we applied a three steps first-layer or pre-processing procedure: (i) Quantitation of inputs (Jia & Rinard, 2020); (ii) Batch normalization (Narodytska et al., 2019b); and (iii) Step Function. Details are presented in Appendix A.3.2.

**Final layer.** TT-DCNN uses a single linear layer as a classifier block. It is straightforward to grasp, even for non-experts and it can be easily encoded into SAT using pseudo-boolean constraint as detailed in Section 2

as long as the weights are integers numbers. As shown in (Jia & Rinard, 2020), we also incorporate batch normalization after the last layer for improving test accuracy. Please refer to Appendix A.3.3 for more details.

### 3.4 Interpretation of our TT-DCNN model and two important properties

**Interpretation.** To illustrate the ease of interpretability of our conversion method, we now offer an interpretation of the SAT formula of the filter with weight matrix $W_1$ defined in the example of Section 3.1: $\mathsf{SAT}_1^{\mathsf{DNF}} = (x_3 \wedge \overline{x_2}) \vee x_0$. First of all, we can observe that the expression is now independent of the value of $x_1$. Then, because the formula is a DNF, we observe that there are two conditions to activate the feature. We can define the set of masks $\mathcal{S}_1 = \{M_1, M_2\}$ with $M_1 = \begin{bmatrix} N & N \\ 1 & 0 \end{bmatrix}$ and $M_2 = \begin{bmatrix} 1 & N \\ N & N \end{bmatrix}$, corresponding to the clause $x_2 \wedge \overline{x_3}$ and $x_0$ respectively, with $N$ denoting `Null`. We can observe that if one of the two masks "matches" the patch of the image, the feature corresponding to the patch position $(i, j)$ will be activated. By "matching", an operation denoted as $\equiv$, we consider that the `Null` position does not matter (i.e. it can be either a 0 or a 1) and that only the 0 and 1 positions should exactly match the 0 and 1 positions of the transformed binary input in order to activate the feature. A formal definition of the operator $\equiv$ is given in Appendix A.4. The features for filter 1 are therefore indicator functions in the case of our model.

More generally, from the $\mathsf{SAT}_f^{\mathsf{DNF}}$ expression we can establish $\mathcal{S}_f$, the set of masks for filter $f$. Then, the value $V$ at position $(i, j)$, for filter $f$ is:

$$V(f, i, j) = \begin{cases} 1 & \text{if } \exists\, M \in \mathcal{S}_f \text{ such that } P(L, i, j) \equiv M \\ 0 & \text{otherwise} \end{cases}$$

In our example with $f = 1$, the feature $V(1, i, j)$ would be activated if $P(L, i, j) \equiv M_1$ or $P(L, i, j) \equiv M_2$. As $V$ is the feature vector before the last classification layer, the result for each class becomes a weighted sum of these indicator functions. For a particular class, these features add up in a weighted way. This means that these sets either participate in favor of the class (positive weight) or against (negative weight) or are independent (weight equals zero). The knowledge of the TT-DCNN therefore resides in $\mathcal{S}_f$. The existence and the analysis of $\mathcal{S}_f$ creates a strong link between XAI and formal verification. To the best of our knowledge, this is the first time that all these masks are formulated and extracted for real-valued DCNNs. The fact that all the masks can be exhaustively given leads to two main properties of TT-DCNN: post-tuning and tractability.

**Two important properties: post-tuning and tractability.** After the training, we can modify the set of learned masks $\mathcal{S}_f$ for a specific purpose. We will see in the next section how we can build a heuristic to characterize overfitting masks, how we can remove them and therefore increase the overall robustness of our model by hand. Another particularity of this architecture is that we can calculate all the possible inputs/outputs of the 2D-CNN block. The Section 4 describes the application of the latter feature to drastically decrease the SAT complexity and therefore enabling the use of a general SAT-solver and an exact model counting solver.

### 3.5 Experiments

**Results.** Our aim is to provide a fully interpretable SAT-convertible model with high natural accuracy. Table 6 compares the performance of a given real-valued DCNN with its corresponding TT-DCNN and BNN (cf. results and architectures in Appendix A.9.1). As expected, this TT-DCNN gives better natural accuracy than BNN (+1.37% for MNIST, +0.57% for CIFAR10) but inferior than the original DCNN (-0.14% for MNIST, -12.05% for CIFAR10). By increasing the truth table dimension $n$ to 27, the gap with the BNN performances increases: +1.49% for MNIST, +4.05% for CIFAR10. Those results show that our network achieves sufficient accuracy for practical use. A summary comparison between BNN, DCNN and TT-DCNN is given in Table 7.

**Limits and further works.** Our results look promising for further scaling up, by adding a block layer or/and increasing the truth table dimension (for $n \geq 10$). We could also attempt using a probability function instead of an indicator function in order to increase interpretability. We could also propose more interpretable sets $\mathcal{S}_f$. For instance, before using the MNIST dataset, we could force training filters with relevant geometrical shapes (arc of circle, circle, line etc.). Ideally, it should be possible to give theoretical masks according to the dataset or to the problem.

**Conclusion.** We have described the TT-DCNN architecture, its encoding and main properties. We then compared our model in the exact verification field as done previously (Narodytska et al. (2018); Narodytska et al. (2019b); Jia & Rinard (2020)). We will show that the TT-DCNN post-tuning and tractability properties are not just theoretical. The first feature addresses the question of how to inject human feedback into a model. The second leads to SAT formulas that can be solved by classic SAT solvers and exact model counting. In doing so, we improve the previous state-of-the-art of SAT complete neural network robustness verification for high and low noise levels.

## 4 APPLICATION TO COMPLETE ROBUSTNESS VERIFICATION

We previously described the TT-DCNN architecture, its encoding and main properties. We have established that our model gives the user the freedom to modify the set of fully explicit masks $\mathcal{S}_f$ for all filters $f$ and to pre-compute all possible outputs of the TT-DCNN before proceeding to production. In this section, we show how to use the first property to address a very general problem: adding domain knowledge into a trained TT-DCNN. Namely, we wish to reduce the model overfitting part in order to increase the verifiable accuracy without modifying the natural accuracy. To do so, we first propose a characterization of the masks responsible for the overfitting of the TT-DCNN followed by a very simple suppression process. Finally, we will see how the tractability property decreases the final layer encoding size. We validate our approach by testing several performance metrics of our model in comparison with other specific state-of-the art models.

### 4.1 POST-TUNING: CHARACTERIZING AND FILTERING OVERFITTING MASKS

**General.** One may envision the set $\mathcal{S}_f$ as a bag full of masks. Upon training, this set is fixed such that it can be considered as the TT-DCNN knowledge. Since we have made these masks explicit (see Section 3.4), we may then modify them for a specific purpose. Drawing on the example of Section 3.4, we can consider that the mask $M_2 = \begin{bmatrix} 1 & N \\ N & N \end{bmatrix}$ is too general and decide to remove it from the set $\mathcal{S}_1$. By doing so, we modify the activation of filter 1 by integrating human knowledge in post-tuning.

**Characterizing overfitting masks.** A mask is considered as an overfitting mask if the ratio of the number of `Null` values over the total number of variables in the mask is below a certain threshold (cf Appendix A.6). Indeed, the less `Null` values in the mask, the more constraints on the input image there are, which hinders the model to generalize properly. The extreme case of the zero ratio means that the filter is only active if the patch exactly overlaps the mask. Therefore, by changing only one bit, we can deactivate that mask.

**Deleting overfitting masks.** First, we remove the overfitting tagged masks defined above with a given `Null` ratio. Then, among the remaining masks, we tag as additional overfitting masks those having the minimum `Null` ratio in the formula. Finally, we apply a random Bernoulli process to partially delete tagged masks. This technique increases the verifiable accuracy of the TT-DCNN models trained on CIFAR10: from 22.79% to 23.08%, without affecting the natural accuracy: from 31.18% to 31.13%. The results are shown in Table 1.

### 4.2 TRACTABILITY: ENCODING THE ADVERSARIAL SETUP INTO SAT

**General.** Another feature of the TT-DCNN's architecture is that one can calculate all the possible inputs/outputs of the 2D-CNN block before using the model in production.

**Encoding pixel noise.** Let us consider some noise (*e.g.* norm-bounded by $l_\infty$) added to the input image prior binarization. After the preprocessing layer, a binarized pixel may either remain unchanged by the perturbation and so fixed as 0 or 1, or it may flip (from 0 to 1 or from 1 to 0) and hence we consider them "unknown", further denoted as $U$. Let us see how this unknown binarized pixel can be integrated in the example of Section 3.4. As there are 4 entries, there are $3^4 = 81$ input possibilities for the block level in an adversial/production setting due to noise: $\begin{bmatrix} 0 & 0 \\ 0 & 0 \end{bmatrix}, \begin{bmatrix} 0 & 0 \\ 0 & 1 \end{bmatrix}, \begin{bmatrix} 0 & 0 \\ 0 & U \end{bmatrix}, \cdots, \begin{bmatrix} U & U \\ U & U \end{bmatrix}$. More generally, as the maximum number of variables for the SAT entry was set to $n = 9$ in this paper, there are therefore a maximum of $3^9 = 19683$ input possibilities for our TT-DCNN . Those unknown binarized pixels are therefore the first literals of our SAT equation: they are the gateway to noise propagation.

**Noise propagation at the block level.** In order to encode noise propagation through blocks, we encode in SAT the input/output relationship $y_f^{(i,j)} = \mathsf{SAT}_f^{\mathsf{CNF}}(x_1^{(i,j)}, \cdots, x_n^{(i,j)})$ for all $(i,j)$ and $f$. For example, for the filter 1 $\mathsf{SAT}_1^{\mathsf{DNF}} = (x_2 \wedge \overline{x_3}) \vee x_0$, the entry $\begin{bmatrix} 0 & 1 \\ 0 & U \end{bmatrix}$ gives $x_3 = U$ as output. The latter result depends on $U$, whereas the input $\begin{bmatrix} 0 & U \\ 0 & 0 \end{bmatrix}$ gives 0 (whatever $U$ is). Hence, there are two noticeable cases: either $U$ propagates through the block (case 1), either it does not (case 2). This example of pre-calculation of inputs/outputs illustrates the superior feature of our model allowing to know precisely how the noise will propagate. This comes in sharp contrast with the currently available BNN model where one has to consider that the noise propagates through the layers in both cases.

**Encoding the attack at the final layer level.** Ultimately, for the final layer, we encode $r_{i,j} = y_i - y_j > 0$ as a reified cardinality constraint which denotes whether the score of class $i$ is higher than the score of class $j$. Being able to distinguish between known and unknown elements of the feature $V$ of the image allows us to reduce significantly the size of the SAT formulas when compared to current state-of-the-art models. Looking at MNIST high noise for instance, our SAT equation yields on average 4K clauses and 1K variables (Table 1). This is a substantial improvement over previously published works where (Jia & Rinard, 2020) has 21K clauses and 48K variables and where (Narodytska et al., 2019b) reported at least 20K clauses and 8K variables. Similar trends were observed with CIFAR10 noise (Table 1). The drastic reduction in the size of SAT formulas renders our model truly amenable to formal verification. Indeed, our SAT verification step is much more tractable for general SAT solvers and exact model counting solvers.

### 4.3 EXPERIMENTS

**Untargeted attack.** Table 1 presents gathered results for natural accuracy, verifiable accuracy for $l_\infty$-norm bounded input perturbations, verification time and the average number of clauses and variables in the SAT formulas on MNIST and CIFAR10. We compare our work with the state-of-the-art of exact verification for BNNs (Jia & Rinard, 2020) and for real-valued networks (Xiao et al. (2018); Tjeng et al. (2017)). We present model architectures details and experimental settings in Appendix A.3.1 and A.6. Table 1 shows that our model after filtering masks considered to be overfitting is more robust. Moreover, our verifiable accuracy is always superior to that of the BNN and even superior to that of the real value based model in the CIFAR10 high noise case. We also show that, with a general SAT solver (here MiniCard (Liffiton & Maglalang, 2012)), we reach a resolution time competitive with BNNs and much lower than the real-valued model. The latter

results, as proposed in (Jia & Rinard, 2020) is mainly due the fact that it is hard to formally verify floating point errors. Thus, we reached our goals: we developed a new a model that is both highly interpretable and competitive with the state-of-the-art. Extended comparison are given in Appendix A.7.

Table 1: Comparison of TT-DCNN with and without filtering with state-of-the-art regarding complete adversarial robustness verification for high noise bounded by $l_\infty$ (results are reported as in the original articles).

| Dataset and Noise Level | Complete Verification Method | | Accuracy | | Mean time (s) | Timeout | #cls/#vars |
|---|---|---|---|---|---|---|---|
| | | | Verifiable | Natural | | | |
| MNIST | SAT-based | TT-DCNN (Ours) | 94.24% | 97.77% | 0.2885 | 0 | 1K/0.4K |
| | | TT-DCNN + Filtering (Ours) | 94.26% | 97.70% | 0.3724 | 0 | 1K/0.4K |
| | | Jia & Rinard (2020) | 91.68% | 97.46% | 0.1115 | 0 | 21K/48K |
| $\epsilon_{test} = 0.1$ | Real-value-based | Xiao et al. (2018) | 94.33% | 98.68% | 5.47 | 0.05% | - |
| | | Tjeng et al. (2017) | 95.62% | 98.11% | 3.52 | 0 | - |
| MNIST | SAT-based | TT-DCNN (Ours) | 79.93% | 96.79 % | 0.4135 | 0 | 4K/1K |
| | | TT-DCNN + Filtering (Ours) | 80.36% | 96.73 % | 0.5722 | 0 | 4K/1K |
| | | Jia & Rinard (2020) | 77.59% | 96.36% | 0.1179 | 0 | 21K/48K |
| $\epsilon_{test} = 0.3$ | Real-value-based | Xiao et al. (2018) | 80.68% | 97.33% | 7.12 | 1.02% | - |
| | | Tjeng et al. (2017) | 74.21% | 86.60% | 5.13 | 0 | - |
| CIFAR10 | SAT-based | TT-DCNN (Ours) | 32.72% | 40.67% | 0.1988 | 0 | 0.7K/0.3K |
| | | TT-DCNN + Filtering (Ours) | 33.04 % | 40.62% | 0.7782 | 0 | 0.7K/0.2K |
| | | Jia & Rinard (2020) | 32.18% | 37.75% | 0.0236 | 0 | 33K/70K |
| $\epsilon_{test} = 2/255$ | Real-value-based | Xiao et al. (2018) | 45.93% | 61.12% | 66.08 | 1.86% | - |
| CIFAR10 | SAT-based | TT-DCNN (Ours) | 22.79% | 31.18% | 0.1635 | 0 | 1K/0.4K |
| | | TT-DCNN + Filtering (Ours) | 23.08% | 31.13% | 0.3887 | 0 | 1K/0.4K |
| | | Jia & Rinard (2020) | 22.55% | 35.00% | 0.1781 | 0 | 9K/13K |
| $\epsilon_{test} = 8/255$ | Real-value-based | Xiao et al. (2018) | 20.27% | 40.45% | 60.67 | 2.47% | - |

**Likelihood of adversarial examples.** The objective of this experiment is to demonstrate that our approach, thanks to its tractability property, allows for the first time the use of exact model counting techniques for low and high noise likelihood estimation. As in (Narodytska et al., 2019b) work, we define the probability that a perturbation is an adversarial example as the number of input perturbations that leads to an attack divided by the total number of input perturbations. For example, for CIFAR10 high noise, the probability of running into an attack is 16.4%, while for low noise it is 10.0%. Complementary results are given in Appendix A.8.

**Limits and further works.** As it is the case with all complete verfication methods, the main limitation of TT-DCNN appears with high noise where observed verifiable and natural accuracies diverge siginificantly and are very far from the computer vision state-of-the art. Further work is required to address this issue. One direction would be to develop a pipeline with exact model counting in order to estimate the prediction distribution of our model at a given noise level.

## 5 CONCLUSION

We presented a novel architecture of SAT encodable real-valued DCNN based on truth tables. This enables a global and exact interpretability as well as post-processing the model. It exhibits competitive performance given low and high noise with state-of-the-art complete verification methods on MNIST and CIFAR10. In a world where DNNs and DCNNs will be widely embedded, the importance of completly verifying the robustness of neural networks to input perturbations is growing. We believe that the TT-DCNN demonstrated robustness constitutes a suitable response to the rising demand for functional formal verification.

## 6 REPRODUCIBILITY

The project code reproducibility for can be found at this URL address[2]

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

## A    APPENDIX

### A.1    COMPLETE VERIFICATION & ROBUSTNESS.

Property verification of SAT-convertible DNNs has been presented in (Narodytska et al., 2018) as follows: given a precondition $prec$ on the inputs $x$, a property $prop$ on the outputs $o$ and a SAT relations given by a DNN between inputs/outputs denoted as $DNN(x, o)$, we check whether the following statement is valid: $prec(x) \land DNN(x, o) \implies prop(o)$. In order to seek a counterexample to this property, we look for a satisfying assignment of $prec(x) \land DNN(x, o) \land \overline{prop(o)}$. An application example of property verification is to check for the existence of adversarial perturbation in a trained DNN. In this case, $prec$ defines an $\epsilon$-ball of valid perturbations and $prop$ states that the classification should not change under small perturbations. Therefore, we distinguish the traditional "natural accuracy" from the "verified accuracy" the later measuring the fraction of the predictions which remains correct for all adversarial attacks within the perturbation constraints.

### A.2    EXAMPLES OF PSEUDO-BOOLEAN CONSTRAINT ENCODING.

The first row of Table 2 presents one inequality example containing 3 natural variables $(x_1, x_2, x_3)$. The corresponding output SAT encoding are given for 5 different published methods: (Abío et al. (2011), Hölldobler et al. (2012) , Eén & Sörensson (2006), Manthey et al. (2014)).

We can see that we systematically end-up with many literals and clauses in the SAT equation. This add complexity to solve the problem. Moreover, there is no straightforward relationship between the SAT formula literals $l_i$ and the inequality variables $x_i$, or between the coefficient of the inequality and the clauses.

Table 2: Examples of inequality conversion into SAT formulas according to different methodologies

| Inequality to convert into SAT Formulas | $x_1 - 2x_2 + 3x_3 \leq 3$ |
|---|---|
| Encoding 1 - Abío et al. (2011) | $(l_4) \wedge (\overline{l_1} \vee l_2 \vee \overline{l_5}) \wedge (l_5 \vee \overline{l_3} \vee \overline{l_6}) \wedge (l_6)$ |
| Encoding 2 - Hölldobler et al. (2012) | $(\overline{l_4} \vee l_9) \wedge (\overline{l_5} \vee l_{10}) \wedge (\overline{l_6} \vee l_{11})$ $\wedge (\overline{l_7} \vee l_{12}) \wedge (\overline{l_8} \vee l_{13}) \wedge (\overline{l_9} \vee l_{14}) \wedge (\overline{l_{10}} \vee l_{15}) \wedge (\overline{l_{11}} \vee l_{16})$ $\wedge (\overline{l_{12}} \vee l_{17}) \wedge (\overline{l_{13}} \vee l_{18}) \wedge$ $(\overline{l_3} \vee l_4) \wedge (\overline{l_3} \vee l_5) \wedge (\overline{l_3} \vee l_6) \wedge$ $(l_2 \vee l_9) \wedge (l_2 \vee l_{10}) \wedge (\overline{l_1} \vee 14) \wedge (\overline{l_4} \vee l_2 \vee l_{11}) \wedge$ $(\overline{l_5} \vee l_2 \vee l_{12}) \wedge (\overline{l_6} \vee l_2 \vee l_{13}) \wedge (\overline{l_9} \vee \overline{l_1} \vee l_{15}) \wedge$ $(\overline{l_{10}} \vee \overline{l_1} \vee l_{16}) \wedge (\overline{l_{11}} \vee \overline{l_1} \vee l_{17}) \wedge (\overline{l_{12}} \vee \overline{l_1} \vee l_{18}) \wedge (\overline{l_7}) \wedge$ $(\overline{l_8}) \wedge (\overline{l_7} \vee l_2) \wedge (\overline{l_{13}} \vee \overline{l_1})$ |
| Encoding 3 - Eén & Sörensson (2006) | $(l_5 \vee \overline{l_3} \vee l_2) \wedge (l_7 \vee \overline{l_3} \vee \overline{l_1}) \wedge (\overline{l_8} \vee \overline{l_3}) \wedge (\overline{l_8} \vee l_2) \wedge (l_6 \vee l_8 \vee \overline{l_7}) \wedge (l_4 \vee \overline{l_6} \vee \overline{l_5}) \wedge (\overline{l_4})$ |
| Encoding 4 - Eén & Sörensson (2006) | $(\overline{l_3} \vee \overline{l_1} \vee \overline{l_4}) \wedge (l_3 \vee l_1 \vee \overline{l_4}) \wedge (\overline{l_3} \vee l_1 \vee l_4) \wedge (l_3 \vee \overline{l_1} \vee l_4) \wedge (l_3 \vee \overline{l_5}) \wedge$ $(l_1 \vee \overline{l_5}) \wedge (\overline{l_3} \vee \overline{l_1} \vee l_5) \wedge (l_3 \vee \overline{l_2} \vee l_5 \vee \overline{l_6}) \wedge (l_3 \vee l_2 \vee \overline{l_5} \vee \overline{l_6}) \wedge$ $(\overline{l_3} \vee \overline{l_2} \vee \overline{l_5} \vee \overline{l_6}) \wedge (\overline{l_3} \vee l_2 \vee l_5 \vee \overline{l_6}) \wedge (\overline{l_3} \vee l_2 \vee \overline{l_5} \vee l_6)$ $\wedge (\overline{l_3} \vee \overline{l_2} \vee l_5 \vee l_6) \wedge (l_3 \vee \overline{l_2} \vee \overline{l_5} \vee l_6) \wedge (\overline{l_2} \vee l_5 \vee \overline{l_7}) \wedge (l_3 \vee l_5 \vee \overline{l_7}) \wedge (l_3 \vee \overline{l_2} \vee \overline{l_7})$ $\wedge (l_2 \vee \overline{l_5} \vee l_7) \wedge (\overline{l_3} \vee \overline{l_5} \vee l_7) \wedge (\overline{l_3} \vee l_2 \vee l_7) \wedge (\overline{l_7} \vee \overline{l_6} \vee l_3) \wedge$ $(\overline{l_7} \vee \overline{l_6} \vee \overline{l_2}) \wedge (\overline{l_7} \vee \overline{l_6} \vee l_5) \wedge (l_7 \vee l_6 \vee \overline{l_3}) \wedge (l_7 \vee l_6 \vee l_2) \wedge (l_7 \vee l_6 \vee \overline{l_5}) \wedge (\overline{l_7} \vee \overline{l_6})$ |
| Encoding 5 - Manthey et al. (2014) | $(l_4) \wedge (\overline{l_3} \vee l_5) \wedge (\overline{l_1} \vee l_5) \wedge (\overline{l_3} \vee \overline{l_1} \vee l_6) \wedge (\overline{l_3} \vee l_7) \wedge (l_2 \vee l_7) \wedge (\overline{l_3} \vee l_2 \vee l_8) \wedge (\overline{l_7} \vee l_9) \wedge$ $(\overline{l_8} \vee l_{10}) \wedge (\overline{l_6} \vee l_9) \wedge (\overline{l_7} \vee \overline{l_6} \vee l_{10}) \wedge (\overline{l_8} \vee \overline{l_6} \vee l_{11}) \wedge (\overline{l_1} 1)$ |

## A.3 MODEL DESCRIPTION

### A.3.1 OVERALL ARCHITECTURE

In this study, we considered the two architectures shown in Table 3. All the paddings are set to 0.

Table 3: Different studied model architectures details.

| Dataset | Name | Layers | Number of Block | Size filters | Kernels | Groups | Strides | Features | Parameters | FLOP | Patch Size |
|---|---|---|---|---|---|---|---|---|---|---|---|
| MNIST | Model Small | 4 | 2 | 60-48-384-48 | 3-1-2-1 | 1-1-24-24 | 3-1-2-1 | 768 | 15488 | 0.63 m | (6,6) |
| | Model Big | 4 | 2 | 60-48-384-48 | 3-1-3-1 | 1-1-48-48 | 2-1-2-1 | 2352 | 32530 | 1.77 m | (7,7) |
| CIFAR 10 | Model Small | 4 | 2 | 60-48-384-48 | 3-1-2-1 | 3-3-24-24 | 3-1-2-1 | 1200 | 17890 | 0.89 m | (6,6) |
| | Model Big | 4 | 2 | 60-48-384-48 | 3-1-3-1 | 3-3-48-48 | 2-1-2-1 | 2352 | 32530 | 1.87 m | (7,7) |

### A.3.2 FIRST LAYER

Like in (Jia & Rinard, 2020), before applying the batch normalisation and the step function, we quantize the inputs as $x^q = \lfloor \frac{x}{q} \rfloor \cdot s$ where $x$ is the real-valued input, $x^q$ is the quantized input to be fed into the TT-DCNN and $s$ is the quantization step size which can be set to $s = 1/255$ to emulate 8-bit fixed-point values, or $2\epsilon$ for adversarial training with a $l_\infty$ disturbance limit of $\epsilon$.

### A.3.3 LAST LAYER

**Last layer.** The last linear layer is composed of a linear layer and a batch normalisation. The weights of the linear layer can be natural instead of binary but this leads to a large increase in the size of the SAT formulas.

**Natural features.** We may also increase the amount of information held by the vector of features $V$ by accepting natural values. Coming back to our previous example, we can see the output $y = [0, -5, 3, -2, 1, -4, 4, -1, 10, 5, 13, 8, 11, 6, 14, 9]$ as $y = -5 \times [0, 1, 0, 0, 0, 0, 0, 0, 0, 0, 0, 0, 0, 0, 0, 0] + 3 \times [0, 0, 1, 0, 0, 0, 0, 0, 0, 0, 0, 0, 0, 0, 0, 0] + \cdots + 9 \times [0, 0, 0, 0, 0, 0, 0, 0, 0, 0, 0, 0, 0, 0, 0, 1]$. Therefore, each of these coefficients can have an associated SAT expression of its own.

For the sake of natural accuracy performance, as in (Evans et al., 2021), it is advisable to encode features and weights on 8 bits. Results for features and weights encoded on 8 bits are given in Appendix A.9. However, the results in this paper are given for binary features and weights.

As we trained the TT-DCNN with a final linear layer then a batch normalisation, we encoded the last layer for the label $i$ as follow:

$$y_i = \sum_{k=1}^{|V|} w_{k,i} V_k + b_i$$

and

$$y_i^{BN} = \texttt{BatchNorm}(y_i) = \gamma \cdot \frac{y_i - \mathbb{E}(y_i)}{\sqrt{Var[y_i] + \epsilon}} + \beta$$

$$y_i^{BN} = \texttt{BatchNorm}(y_i) = \frac{\sum_{k=1}^{|V|} \gamma w_{k,i} V_k + \gamma b_i - \mathbb{E}(y_i)}{\sqrt{Var[x] + \epsilon}} + \beta$$

with $\epsilon = 1e - 5$ and $\cdot$ denotes element-wise multiplication.

$$y_i^{BN} = \sum_{k=1}^{|V|} \tilde{w}_{k,i} V_k + \tilde{b}_i$$

with

$$\tilde{w}_{k,i} = \lfloor \frac{\gamma w_{k,i}}{\sqrt{Var[y_i] + \epsilon}} \rfloor$$

$$\tilde{b}_i = \frac{\gamma b_i - \mathbb{E}(y_i)}{\sqrt{Var[y_i] + \epsilon}} + \beta$$

To facilitate the SAT conversion, we also restrict the variance statistics and the scale parameter $\gamma$ in Batch-Norm() of the last layer to be scalars computed on the whole feature map rather than per-channel statistics. Before rounding, we multiply $\tilde{w}_{k,i}$ and $\tilde{b}_i$ by 100 in order to keep some details contained in the floating points.

### A.4 OPERATOR

We note as $\cdot$ the operator standing for the product term by term between two matrices. A mask (for example $M_1 = \begin{bmatrix} N & N \\ 1 & 0 \end{bmatrix}$) can be decomposed into two masks: $M^{(0)}$ and $M^{(1)}$. $M^{(0)}$ is 0 where $M$ takes the value $\texttt{Null}$, 1 everywhere else (ie as $M_1^{(0)} = \begin{bmatrix} 0 & 0 \\ 1 & 1 \end{bmatrix}$). $M^{(1)}$ is 1 where $M$ takes the value 1, 0 everywhere else (ie as $M_1^{(1)} = \begin{bmatrix} 0 & 0 \\ 1 & 0 \end{bmatrix}$).

Now let us note $I$ the input matrices of the 2D-CNN and $\tilde{M}$ the matrix given by $\tilde{M} = I \cdot M^{(0)}$. Then, we say that $M$ matches $I$ if and only if: $\forall(i,j) \ \tilde{M}[i,j] = M^{(1)}[i,j]$.

## A.5 Details of TT-DCNN encoding

### A.5.1 First layer

Published as in Narodytska et al. (2019b), we note $x_q$ the quantified pixel variable, and $x_b$ the bit variable after the first preprocessing layer, we have $x_b = sign(\frac{\alpha}{\sigma}(x_q - \mu) + \gamma) = sign(x_{inter})$ with $\alpha$, $\sigma$, $\mu$, $\gamma$ the characteristic of the first batch normalisation. We consider two extreme values of the expression that is an input of sign w.r.t. $\epsilon$. We have (for $\alpha > 0$) two extreme point: $x_{inter}^{max} = \frac{\alpha}{\sigma}(x_q + \epsilon - \mu) + \gamma$ and $x_{inter}^{min} = \frac{\alpha}{\sigma}(x_q - \epsilon - \mu) + \gamma$

If $x_{inter}^{min} \geq 0$ then we know that $x_b = 1$. If $x_{inter}^{max} < 0$ then we know that $x_b = 0$. Otherwise, $x_b \in \{0, 1\}$, therefore it's a literal for the SAT encoding. We can consider the reverse transformation. If $x_b = 0$ (i.e. for $x_b = 1$) is a solution of a problem produced by the SAT solver then we can map back to $x_{attack} = x - \epsilon$ (i.e. for $x_{attack} = x + \epsilon$).

### A.5.2 Last Layer

Let $V$ be the feature vector before the last linear layer, and $\mathcal{U}$ the set of indices of $V$ that are unknown due to noise propagation and $\mathcal{K}$ those that are known. We have, for the class attack $a$ higher than class target $t$:

$$\sum_{k=1}^{|V|} w_{k,a} V_k > \sum_{l=1}^{|V|} w_{l,t} V_l$$

We can divide the set $[1 \cdots n]$ into $\mathcal{U}$ and $\mathcal{K}$

$$\sum_{k \in \mathcal{K}} w_{k,a} V_k + \sum_{k \in \mathcal{U}} w_{k,a} V_k > \sum_{l \in \mathcal{K}} w_{l,t} V_k + \sum_{l \in \mathcal{U}} w_{l,t} V_l$$

We group together the set known and

$$\sum_{k \in \mathcal{U}} (w_{k,t} - w_{k,a}) V_k < \sum_{l \in \mathcal{K}} (w_{l,a} - w_{l,t}) V_l$$

We note $c = \sum_{l \in \mathcal{K}} (w_{l,a} - w_{l,t}) V_l$, then $c$ is a integer number. We have

$$\sum_{k \in \mathcal{U}} (w_{k,t} - w_{k,a}) V_k \leq c - 1$$

We note $G = \text{GCD}(abs(w_{k,t} - w_{k,a}) \forall k \in \mathcal{U})$, then we have:

$$\sum_{k \in \mathcal{U}} \frac{(w_{k,t} - w_{k,a})}{G} V_k \leq \frac{c-1}{G}$$

As $\frac{c-1}{G}$ need to be an integer, we have:

$$\sum_{k \in \mathcal{U}} \frac{(w_{k,t} - w_{k,a})}{G} V_k \leq \lfloor \frac{c-1}{G} \rfloor$$

This inequality is encoded with the project (Ignatiev et al., 2018).

## A.6 EXPERIMENTAL DETAILS

**Experimental environment.**   The project was implemented with Python and the library PyTorch (Paszke et al., 2019). The project code can be found at this URL address[3]. Our work station is constituted of a GPU Nvidia GeForce GTX 970 with 4043 MiB memory and four Intel core i5-4460 processors clocked at 3.20 GHz.

**Training method.**   We build our training method on the top of Jia & Rinard (2020) project and we refer to their notations for this section. We train the networks using the Adam optimizer (Kingma & Ba, 2014) for 90 epochs with a minibatch size of 128. The mean and variance statistics of batch normalization layers are recomputed on the whole training set after training finishes.

Learning rate is 0.0005. We use PGD with adaptive gradient cancelling to train robust networks, where the perturbation bound $\epsilon$ is increased linearly from 0 to the desired value in the first 50 epochs and the number of PGD iteration steps grows linearly from 0 to 10 in the first 23 epochs.

The parameter $\alpha$ in adaptive gradient cancelling is chosen to maximize the PGD attack success rate evaluated on 40 minibatches of training data sampled at the first epoch. Candidate values of $\alpha$ are between 0.6 to 3.0 with a step of 0.4. Note that $\alpha$ is a global parameter shared by all neurons. We do not use any data augmentation techniques for training. Due to limited computing resource and significant differences between the settings we considered, data in this paper are reported based on one evaluation run.

**Weight initialization.**   Weights for the final connected layers are initialized from a Gaussian distribution with standard deviation 0.01, and the mask weights in BinMask are enforced to be positive by taking the absolute value during initialization.

**Other hyperparameters.**   The input quantization step s is set to be $0.61 = 0.3 \times 2 + 0.01$ for training on the MNIST dataset, and $0.064 \approx 16.3/255$ for CIFAR10, which are chosen to be slightly greater than twice the largest perturbation bound we consider for each dataset. Except for CIFAR10 for wich we double the trainig noise, the training noise level is equal to the testing noise level. The CBD loss is applied on MNIST high noise only and $\nu$ is set to be $5e-4$, 0 otherwise. We apply a weight decay of $1e-7$ on the binarized mask weight of BinMask. We use the encoding proposed in (Abío et al., 2011). In Table 1, we use model Big except for the high noise model with CIFAR10. For the post tunning parameters, we use a proportion ratio of 0.1 and a probability $p = 0.05$ for MNIST low noise, we double it for high noise and a proportion ratio of 0.05 and a probability $p = 0.01$ for CIFAR10 low noise and we double it for high noise.

## A.7 COMPARISON WITH SAT METHOD FOR EXACT SAME ARCHITECTURE AND EXACT SAME TRAINING CONDITIONS

As the architectures in (Jia & Rinard, 2020) are different from ours, we reproduce the results for BNN with our exact same architecture and the exact same training conditions. We also add an experiment with their model architecture and our training condition for CIFAR10 high noise as we trained it for a noise of 16/255. Results can be found in Table 4. We can observe that our model always outperforms the BNN model in term of verifiable accuracy.

---

[3]https://github.com/iclr2022anonymous/Truth-Table-Deep-Convolutional-Neural-Network

Table 4: Performance of studied models as BNN for equitable comparaison

| Dataset and Noise Level | Model & Training condition | Accuracy | | Mean time (s) | Timeout | #cls/#vars |
|---|---|---|---|---|---|---|
| | | Verifiable | Natural | | | |
| MNIST 0.3 | Model Big as BNN | 49.53% | 93.62% | 0.010 | 0 | 23K/33K |
| CIFAR10 2/255 | Model Big as BNN | 28.43% | 37.66% | 0.008 | 0 | 32K/46K |
| CIFAR10 8/255 | Model Small as BNN | 20.21% | 29.38% | 0.005 | 0 | 15K/22K |
| | Architecture proposed in Jia & Rinard (2020) as BNN with our training noise conditions | 19.86% | 31.95% | 0.002 | 0 | 31K/64K |

## A.8 MODEL COUNTING

**Exact model counting.** Given a CNF formula $\Phi$, the problem of model counting is to calculate the corresponding number of satisfying assignments $\#\Phi$. This problem is complete for the complexity class $\#\mathcal{P}$ (Toda, 1991). A number of tools for exact model counting have been developed and for this study we are using the recent Ganak model counter (Sharma et al., 2019). An application example of model counting is to establish how many adversarial attacks exist for a trained DNN. In this case, $prec$ defines an $\epsilon$-ball of valid perturbations and $prop$ states that the classification should change under small perturbations. And the problem is given to a model counting solver (instead of a SAT solver) in the form of CNF.

**Exact model counting set up.** We encode as condition the input noise and the noise propagation as before. We change the final inequality: by encoding such that the class $i$ is greater than all others. Then, we start the MaxSAT solver on this formula: it gives us how many entries lead to an output if class $i$. We also add a scenario: we randomly fixed 50% of the first layer literals to a random value (0 or 1). This leads to a partial distribution of prediction instead of an exact distribution prediction. We reports $P(adv)$, the probability to get an adversarial, the number of timout of the exact model counting and finaly, the accuracy given by the argmax of the distribution as prediction. The same is given in the noise scenario. We tested the first 1K samples of the dataset for model with and without filtering

**Results.** We performed the experiments for MNIST only. Results can be found in Table 5. First, we observe that, for the Small model, the number of solved samples is very high: almost 100% except in 3 cases: when the final linear is not binary (Ter) and when the training noise is not in line with with the testing noise. This contrasts with the results in Narodytska et al. (2019b) where there is only a single configuration with 100% of the instances solved with an approximated model counting. In the case of Small model, the probability to get an adversarial is pretty high. The latter surprisingly increases with the filtering and with the robust training. On the contrary, for the Big model, the probability to get an adversarial is pretty low and the number of time out increases. However, in the noise setting, the number of timeout drastically decreases, but the probability to get an adversarial also increases. We also highlight the very high accuracy if we return the argmax of the output distribution as prediction. We think these observations can lead to interesting further works.

## A.9 ABLATION STUDY

### A.9.1 NATURAL ACCURACY

We compare the BNNs, TT-DCNN models and the standard real-valued models. We use the two-block Big model, without the amplification blocks for BNN and real-valued model. TT-DCNN has a final layer with

Table 5: Results of TT-DCNN in the likelihood adversarial examples set-up for MNIST

| Model | Noise train | Noise test | Loss type | Amplification | Final Linear | P(adv) | | Accuracy - P(adv) | | Timeout - P(adv) | | P(adv) with 50% noise | | Accuracy - P(adv) with 50% noise | | Timeout - P(adv) with 50% noise | |
|---|---|---|---|---|---|---|---|---|---|---|---|---|---|---|---|---|---|
| | | | | | | Normal | Filtered | Normal | Filtered | Normal | Filtered | Normal | Filtered | Normal | Filtered | Normal | Filtered |
| Small | 0 | 0.3 | 0 | Small | Bin | 0.0119 | - | 716 | - | 101 | - | 0.05 | - | 875 | - | 0 | - |
| | | 0.3 | 0 | Normal | Bin | 0.00084 | 0.00162 | 526 | 549 | 432 | 388 | 0.0135 | 0.012 | 898 | 862 | 13 | 6 |
| | 0.1 | 0.1 | 0 | Normal | Bin | 0.2135 | 0.2194 | 936 | 932 | 0 | 0 | 0.3787 | 0.3559 | 938 | 930 | 0 | 0 |
| | 0.3 | 0.3 | 0 | Normal | Bin | 0.09265 | 0.1068 | 900 | 883 | 1 | 1 | 0.2849 | 0.28267 | 899 | 884 | 0 | 0 |
| | | 0.3 | 3 | Normal | Bin | 0.1003 | 0.1085 | 901 | 863 | 0 | 0 | 0.3019 | 0.289 | 903 | 865 | 0 | 0 |
| | 0.4 | 0.3 | 0 | Small | Bin | 0.4045 | - | 837 | - | 0 | - | 0.467 | - | 832 | - | 0 | - |
| | | 0.3 | 0 | Normal | Bin | 0.2099 | - | 877 | - | 0 | - | 0.3439 | - | 853 | - | 0 | - |
| | | 0.3 | 1 | Normal | Bin | 0.1643 | - | 880 | - | 0 | - | 0.3038 | - | 883 | - | 0 | - |
| | | 0.3 | 3 | Normal | Bin | 0.1914 | 0.2060 | 886 | 877 | 0 | 0 | 0.36322 | 0.3412 | 899 | 898 | 0 | 0 |
| | | 0.3 | 0 | Normal | Ter | 0.0 | - | 641 | - | 354 | - | 0 | - | 795 | - | 195 | - |
| Big | 0 | 0.3 | 0 | Normal | Bin | 0.0 | - | 111 | - | 888 | - | - | - | - | - | - | - |
| | 0.1 | 0.1 | 0 | Normal | Bin | 0.022 | 0.039 | 973 | 970 | 13 | 17 | 0.03 | 0.05 | 979 | 978 | 0 | 1 |
| | 0.3 | 0.3 | 0 | Normal | Bin | 2E-10 | 2E-10 | 802 | 815 | 184 | 174 | - | - | - | - | - | - |
| | | 0.3 | 3 | Normal | Bin | 3E-06 | 2E-06 | 802 | 805 | 196 | 195 | 0.014 | 0.017 | 936 | 942 | 31 | 31 |
| | 0.4 | 0.3 | 0 | Normal | Bin | 0.0053 | - | 864 | - | 108 | - | - | - | - | - | - | - |
| | | 0.3 | 1 | Normal | Bin | 0.00464 | - | 862 | - | 125 | - | - | - | - | - | - | - |
| | | 0.3 | 3 | Normal | Bin | 0.005 | 0.009 | 867 | 864 | 105 | 94 | 0.012 | 0.0377 | 936 | 923 | 8 | 4 |

integer values. We stopped the training at 45 epochs for all three models to prevent overfitting. We also compare TT-DCNN for two values of truth table size $n$: 9 and 27. The results are given in Table 6. We observe that for MNIST and CIFAR10, TT-DCNN outperforms BNN and is inferior to the real-valued model. We also observe that as $n$ increases, TT-DCNN tends to the real-valued model performances. We use $n = 27$ as it seems reasonable to compute $2^{27}$ operations to get the SAT formula of a filter.

Table 6: Comparison of natural accuracy between TT-DCNN, BNN and DCNN.

| | Binary based model BNN | Real based model DCNN | TT-DCNN Our (n = 9) | TT-DCNN Our (n = 27) |
|---|---|---|---|---|
| Natural Accuracy on MNIST | 96.98% | 98.49% | 98.35% | 98.47% |
| Natural Accuracy on CIFAR10 | 53.53% | 66.16% | 54.11% | 58.08% |

Table 7: Comparison of model architectures between TT-DCNN, BNN and DCNN.

| | BNN | DCNN | TT-DCNN |
|---|---|---|---|
| Type weigths | Binary | Floating | Floating |
| Type Intermediate values | Binary | Floating | Mixed Floating - Binary |
| Group CNN | Not Grouped | Not Grouped | Grouped |
| Final linear classification | MLP (Multi-Layer-Perceptron) | MLP (Multi-Layer-Perceptron) | Linear |

### A.9.2 STUDY ON INFLUENCE OF AMPLIFICATION LAYER

We tested three models: a model without amplification (one block is one layer), a model with minimal amplification (one block is two layers - with amplification factor 1) and a model with standard amplification (model presented in Appendix A.3.1). Results are presented in Table 8. We respect the training conditions proposed in Appendix A.6.

We can observe that the amplification layer is always profitable for the natural accuracy.

Table 8: Comparison of the natural accuracy of TT-DCNN for three different types of amplification configuration, for CIFAR10 and MNIST, in the case of training with noise and without noise.

| Dataset | Noise Training | Amplification | Accuracy | |
|---|---|---|---|---|
| | | | Model Small | Model Big |
| MNIST | 0 | No | 89.71% | 95.28% |
| | | Small | 91.66% | 95.79% |
| | | Normal | 94.60% | 97.53% |
| | 0.4 | No | 82.63% | 92.42% |
| | | Small | 85.67% | 94.19% |
| | | Normal | 87.59% | 95.77% |
| CIFAR10 | 0 | No | 40.53% | 44.33% |
| | | Small | 42.51% | 47.12% |
| | | Normal | 44.05% | 50.27% |
| | 16/255 | No | 22.40% | 28.05% |
| | | Small | 27.86% | 37.29% |
| | | Normal | 32.19% | 47.12% |

### A.9.3 OTHERS

For the study, we tested the first 1K samples of the dataset for model small and big with and without filtering. We reported the natural accuracy, the verified accuracy, the total time to compute, the number of clauses and variables. We don't report the number of timeout as they are all 0. The results were computed on an other computer than those of Table 1. We took the measures for different:

- **Size of the model.** We use the two models presented above.
- **Dataset.** We use MNIST and CIFAR10
- **Training noise level.**
- **Testing noise level.**
- **Loss coefficient as introduced in Jia & Rinard (2020).**
- **Amplification Layer.** Two amplifications possibles described in Appendix A.9.2
- **Final Layer.** Two possibilities: with binary (Bin) or ternary (Ter) weigths.

First, we saw that MNIST and CIFAR10 can be scaled to our two models for low and high noise. Then, we observe that the loss coefficient has little impact on the performances. We can highlight that the training noise level has an important impact on the verification accuracy and the time computation. We can observe that most of the time the filtering leads to a better verifiable accuracy and sometimes to a better natural accuracy.

Table 9: Results of TT-DCNN in the untargeted attack set-up for CIFAR10

| Model | Noise train | Noise test | Loss type | Amplification | Final Linear | Natural Accuracy | | Verifiable Accuracy | | Mean time (s) | | #cls/#vars | |
|---|---|---|---|---|---|---|---|---|---|---|---|---|---|
| | | | | | | Normal | Filtered | Normal | Filtered | Normal | Filtered | Normal | Filtered |
| Small | 0 | 8/255 | 0 | Small | Bin | 448 | - | 20 | - | 0.254 | - | 9722/1610 | - |
| | | | 0 | Normal | Bin | 456 | - | 18 | - | 0.266 | - | 10859/1555 | - |
| | 2/255 | 2/255 | 0 | Normal | Bin | 460 | 456 463 | 328 | 330 (light) 299 (strong) | 0.105 | 0.208 0.200 | 661/258 | 639/250 584/233 |
| | | | 3 | Normal | Bin | 463 | 446 | 300 | 288 | 01:45 | 03:25 | 696/273 | 675/269 |
| | 2.2/255 | | 0 | Normal | Bin | 458 | 431 | 305 | 316 | 0.106 | 0.217 | 730/279 | 676/257 |
| | 8/255 | 8/255 | 0 | Small | Bin | 348 | 248 | 190 | 185 | 0.090 | 0.239 | 1349/395 | 1273/390 |
| | | | 0 | Normal | Bin | 357 | 357 356 | 196 | 229 (light) 231 (Strong) | 0.109 | 0.208 0.213 | 2269/509 | 2246/508 2176/507 |
| | | | 1 | Normal | Bin | 368 | 347 | 173 | 186 | 0.110 | 0.223 | 1890/494 | 1519/464 |
| | | | 3 | Normal | Bin | 358 | 299 | 178 | 192 | 0.113 | 0.200 | 2337/557 | 2031/538 |
| | 16/255 | 8/255 | 0 | Small | Bin | 232 | - | 135 | - | 0.061 | - | 1148/368 | - |
| | | | 0 | Normal | Bin | 329 | - | 219 | - | 0.092 | - | 1342/425 | - |
| | | | 1 | Normal | Bin | 344 | - | 194 | - | 0.095 | - | 1339/413 | - |
| | | | 3 | Normal | Bin | 334 | 334 | 194 | 193 | 0.093 | 0.206 | 1473/461 | 1273/428 |
| | | | 0 | Normal | Ter | 390 | - | 139 | - | 0.104 | - | 3119/1302 | - |
| | 16.7/255 | | 0 | Normal | Bin | 347 | 332 | 201 | 244 | 0.998 | 0.199 | 1379/451 | 1300/440 |
| Big | 0 | 8/255 | 0 | Small | Bin | 506 | - | 15 | - | 0.703 | - | 26183/3545 | - |
| | | | 0 | Normal | Bin | 539 | - | 10 | - | 0.640 | - | 23777/3874 | - |
| | 2/255 | 2/255 | 0 | Normal | Bin | 458 | - | 290 | - | 0.158 | - | 1327/452 | - |
| | 8/255 | 8/255 | 0 | Normal | Bin | 380 | - | 153 | - | 0.172 | - | 4099/730 | - |
| | 16/255 | 8/255 | 0 | Small | Bin | 294 | - | 226 | - | 0.193 | - | 3595/813 | - |
| | | | 0 | Normal | Bin | 366 | - | 214 | - | 0.205 | - | 4666/941 | - |
| | | | 0 | Normal | Ter | 389 | - | 88 | - | 0.512 | - | 22757/9828 | - |
| | | | 1 | Normal | Bin | 348 | - | 212 | - | 0.175 | - | 3625/912 | - |
| | | | 3 | Normal | Bin | 327 | - | 108 | - | 0.174 | - | 4661/957 | - |

Table 10: Results of TT-DCNN in the untargeted attack set-up for MNIST

| Model | Noise train | Noise test | Loss type | Amplification | Final Linear | Natural Accuracy | | Verifiable Accuracy | | Mean time (s) | | Timeout | | #cls/#vars | |
|---|---|---|---|---|---|---|---|---|---|---|---|---|---|---|---|
| | | | | | | Normal | Filtered | Normal | Filtered | Normal | Filtered | Normal | Filtered | Normal | Filtered |
| Small | 0 | 0.3 | 0 | Small | Bin | 909 | - | 226 | - | 0.183 | - | 0 | - | 2771 / 795 | - |
| | | | 0 | Normal | Bin | 936 | 908 | 169 | 161 | 0.239 | 0.268 | 0 | 0 | 3209/871 | 2974/838 |
| | 0.1 | 0.1 | 0 | Normal | Bin | 943 | 937 | 848 | 665 | 0.073 | 0.128 | 0 | 0 | 456/199 | 442/195 |
| | 0.3 | 0.3 | 0 | Normal | Bin | 906 | 902 | 674 | 648 | 0.139 | 0.207 | 0 | 0 | 1406/501 | 1370/498 |
| | | | 3 | Normal | Bin | 912 | 866 | 665 | 637 | 0.147 | 0.217 | 0 | 0 | 665 1403 | 1332 493 |
| | 0.4 | 0.3 | 0 | Small | Bin | 819 | - | 670 | - | 0.114 | - | 0 | - | 977/397 | - |
| | | | 0 | Normal | Bin | 859 | - | 676 | - | 0.127 | - | 0 | - | 1091/425 | - |
| | | | 1 | Normal | Bin | 889 | - | 651 | - | 0.146 | - | 0 | - | 1239/467 | - |
| | | | 3 | Normal | Bin | 886 | 881 | 677 | 680 | 0.117 | 0.245 | 0 | 0 | 1059 421 | 1041 419 |
| | | | 0 | Normal | Ter | 930 | - | 641 | - | 0.324 | - | 0 | - | 4016/1887 | - |
| Big | 0 | 0.3 | 0 | Normal | Bin | 974 | - | 111 | - | 0.716 | - | 0 | - | 9879/2637 | - |
| | 0.1 | 0.1 | 0 | Normal | Bin | 976 | 978 977 | 944 | 947 (low) 946 (low) | 0.302 | 0.494 0.491 | 0 | 0 | 1146 461 | 1137 459 1138 459 |
| | 0.3 | 0.3 | 0 | Normal | Bin | 957 | 953 | 771 | 776 | 0.463 | 0.634 | 0 | 0 | 4047/1250 | 3834/1206 |
| | | | 3 | Normal | Bin | 965 | 964 959 | 774 | 778 (high) 773 (high) | 0.485 | 0.634 0.696 | 0 | 0 | 3923 1240 | 3863/1231 3820 1223 |
| | 0.4 | 0.3 | 0 | Normal | Bin | 951 | - | 790 | - | 0.423 | - | 0 | - | 3163/1039 | - |
| | | | 1 | Normal | Bin | 957 | - | 808 | - | 0.446 | - | 0 | - | 3117 1062 | - |
| | | | 3 | Normal | Bin | 954 | 931 | 808 | 781 | 0.436 | 0.590 | 0 | 0 | 2816 980 | |
| | | | 0 | Normal | Ter | 974 | - | 723 | - | 10,44 | - | 0 | - | - | |

