# OpenReview forum: "Truth Table Deep Convolutional Neural Network, A New SAT-Encodable Architecture - Application To Complete Robustness"
_ICLR.cc/2022/Conference — ICLR 2022 Submitted_

### Official Review · Reviewer_UdYz · 2021-10-27

**Correctness:** 2
**Technical Novelty And Significance:** 3
**Empirical Novelty And Significance:** 2
**Recommendation:** 5
**Confidence:** 3

**Main Review:**

Explainability and formal verification of deep neural networks are vibrant research venues.
Instead of focusing on techniques for interpreting/verifying the existing models, this work proposes a novel architecture that admits a more compact symbolic encoding *by construction*, resulting in networks that are more easy to interpret and verify. I think that the idea is sensible.

I think that I got the intuitive idea behind TT-DCNs, but the presentation of the architecture could be largely improved, possibly by adding figures and more examples. The core idea seems simple: reducing the ariety of certain functions, in order to exhaustively enumerate input-output pairs in truth tables. This of course comes at a cost in terms of expressiveness, and the trade-off between interpretability and predictive performance is not very clear. Generating abstractions of logical theories is a venerable field of study and the existing techniques can be applied to other models too. The claim that these architectures are more interpretable (and can be fine-tuned by hand) thanks to their truth tables is not very convincing and should be substantiated with experiments, comparing different XAI approaches. With the current version, I am left wondering if interpretability is a natural by-product of the model's limitation to small truth tables.

The writing seems tailored towards experts in this area. The reader is assumed to be knowledgeable about formal verification of neural networks. For instance, "verifiable accuracy" or "complete robustness" are not defined anywhere in the text.

Minors:
The definition of patch function is unclear to me.
"We have P (1) = k and P (s + 1) = P (s) if and only if the kernel size of the layer is 1."
What if the kernel size isn't 1?

"In fact, the feature associated to the filter f is equal to 1 if there exists a mask M in the set of masks S f that matches the input patch, 0 otherwise (we will define later what are a mask, a set of masks and the matching operator)."
I would then avoid mentioning these concepts.

"To the best of our knowledge, this is the first time that the exact knowledge of a real-valued DCNN is formally extracted."
Isn't this true for BNNs compiled to propositional formulas too?


**Summary Of The Paper:**

This article proposes a novel convolutional architecture, dubbed Truth Table DCNs (TT-DCNs). Compared to Binary Neural Networks (BNNs), the proposed architecture admits a much more compact propositional logic encoding while improving their accuracy.
To the best of my understanding, the core idea is using real-weighted convolutions and aggregations that are then binarized with step-functions. These convolutions can be fully represented with truth tables, as long as the number of inputs in the filter is low-dimensional.  Furthermore, the authors claim that the proposed architecture is intrinsically more interpretable than BNNs and that it can be post-tuned by integrating human knowledge in the truth tables.  Experiments show that TT-DCNs are more easily verified while retaining practical predictive
performance.

**Summary Of The Review:**

- The paper addresses a very relevant problem in AI
- Developing deep architectures that are verifiable and interpretable by construction is a sensible idea in my opinion
- The presentation of the proposed architecture could be improved by adding figures and more examples
- Adding some definitions would benefit a large audience of non-experts in verification
- The trade-off between interpretability and predictive performance is not well characterized
- The claim that the model is more interpretable should be substantiated with experiments comparing TT-DCNs with other XAI methods

---

> ### Author Response · Authors · 2021-11-15
> **We would like to thank the reviewer for his detailed comments on our submission. We will try to answer every point you highlight step by step.**
>
>
>
>
>  > "I think that [...] and more examples."
>
>  > " I would t[...] concepts."
>
> The editorial comments will be taken into account in the next version.
>
>
>  > "trade-off [...] clear."
>
>  Sorry, we are not able to understand this comment. Our work is not a trade-off between interpretability and verifiable accuracy, it is a trade-off between verifiable accuracy and verifiable time.
>
>   > "Generating [...] models too."
>
> Currently, in the field of DNN to SAT conversion, there is only one method described in [2]. Section 2,paragraph SAT encoding of neural networks.This is why we believe our paper is interesting:**it proposes a new and second approach to convert DNNs into SAT formulas.**
>
>
>  > "The claim that [...] experiments"
>
> By interpretability, in this paper, we mean a global and exact method. This is the case for our model, as well as for the BNNs.
>
> However, our method is more interpretable than the BNNs verification method because, in our case, we understand what exactly is a literal and a clause for a DNF formula. This is the whole section 3.4 as well as the video (5:53 to 8:15).
>
> Whereas for BNNs, you have to apply the cardinality constraint, which is a very opaque operation. An example of  conversion is given in Table 2: for encoding x1 - 2 x2 + 3 x3 <= 3, the  CNF is (l_4) \land (\overline{l_1} \lor l_2 \lor \overline{l_5}) \land (l_5 \lor \overline{l_3} \lor \overline{l_6}) \land (l_6). Whereas we can make sense of the inequality: the filter of size 2x2: [[1,-2][3,0]] with a bias of -3, then a ReLu function gives the inequality above, if you consider the input [[x1, x2][x3,x4]]. But then, when you turn it into a CNF formula, we can't make sense of either the literals or the clauses. What does l_4 represent in relation to my inputs and filter? Same question for the clause (\overline{l_1} \lor l_2 \lor \overline{l_5})
>
> Therefore, when we incorporate this into the sat solver, one is able to verify the DNN but not to make sense of the SAT equation. In our case, we know: each literal is an input variable and a DNF clause, a mask that has to match the input to activate. **This is why we claim that our method is more interpretable.**
>
>  Moreover, we designed a heuristic to characterize what overfitting is in our case in a logical way. The characteristics of overfitting (proposed in subsection 4.1, paragraph Characterizing overfitting masks) makes sense to everyone and is formally concretised in the case of the filters. Finally, we implemented it .The verifiable accuracy is better with filtering than before: 79.93% to 80.36% for MNIST high noise. All this is not possible in the case of BNN.
>
>
>
>  > "[...] comparing different XAI approaches."
>
> First, our paper is not an interpretability paper but a formal verification paper.However, from what we could observe, XAI methods do not scale on DNNs and image datasets. If we take as example paper [1]:  "Concretely, let us consider a neural network with one hidden layer containing **15 or 20 neurons** trained to distinguish two digits[...]"
>
>
>
> Since our paper just compares our interpretability with that of BNNs, and since XAI methods do not scale  we preferred not to extend to XAIs. However, if the reviewer has suggestions for specific methods, - we happy to experiment.
>
>
>  > "With the [..] truth tables."
>
> As reviewer 1 points out, if the truth table is larger than 48, then the enumeration is not tractable. But below that, our interpretability method is transparent to the size of truth table.
>
>
>  > "For instance, [...] text."
>
> Verifiable accuracy is defined in Appendix A1, last sentence. We realize that we forgot to define complete robustness and this will be added in the next version of our article
>
>  > " What if the kernel size isn't 1?"
>
> In this case, the size of the kernel will also depend on the stride of the previous layer.We will mention this in the next version of our paper.
>
>
>  > " Isn't t[...] formulas too?"
>
>  Only BNNs can be transformed into SAT at this time. In a way what you say is right: {-1,0,1} are particular reals but our method generalizes to any kind of real weight. We refer the reviewer to works [2] and [3] for the conversion to SAT.
>
>
>
>
>
>
> ## Conclusion
>
> **Finally, we would like to recall that we outperform the NeurIPS 2020 paper [1] on verified accuracy: +2.8% for MNIST high noise, +0.5 CIFAR10 high noise (for the same computation time despite the fact that we use a general solver in our case versus a specific one in their case)**
>
> We deplore that this work, which focuses on improving the trade-off of verifiable accuracy and verification time while bringing an interpretative dimension to the whole, was mainly seen by the reviewer as interpretable AI paper as it is not.
>
>
> [1] Abduction-Based Explanations for Machine Learning Models,Ignatievet al.(AAAI2018)
>
> [2] In search for a sat-friendly binarized neural network architecture,Narodytskaetal. (ICLR 2020)
>
> [3] Efficient exact verification of binarized neural networks,Jia&Rinard(NeurIPS 2020)

---

> > ### Comment · Reviewer_UdYz · 2021-11-17
> > **Comments on the authors' feedback**
> >
> > I thank the authors for the detailed answer.
> >
> > > We deplore that this work, which focuses on improving the
> > > trade-off of verifiable accuracy and verification time while
> > > bringing an interpretative dimension to the whole, was mainly
> > > seen by the reviewer as interpretable AI paper as it is not.
> >
> > My apologies, it was not my intention to disregard your results on
> > verifiable accuracy my the review.
> >
> > > our paper is not an interpretability paper but a formal
> > > verification paper.
> >
> > I got a different idea when reading your submission. If this work is
> > purely concerned with formal verification, then I would suggest to
> > rephrase/remove the (numerous) interpretability claims, which are not
> > substantiated in the experimental section.
> >
> > Some examples:
> >
> > > The TT-DCNN architecture enables for the first time all the logical
> > > classification rules to be extracted from a performant neural
> > > network which can be then easily interpreted by anyone familiar with
> > > the domain.
> >
> > > We believe our new architecture paves the way between eXplainability
> > > AI (XAI) and formal verification.
> >
> > > Our aim is to provide a fully interpretable SAT-convertible model
> > > with high natural accuracy.
> >
> > ---
> >
> > > Sorry, we are not able to understand this comment. Our work is not a
> > > trade-off between interpretability and verifiable accuracy, it is a
> > > trade-off between verifiable accuracy and verifiable time.
> >
> > I understand that there is a trade-off between accuracy and
> > verification time. The more expressive (more layers, less grouping,
> > larger truth tables, ...) the model, the harder it is to verify. Isn't
> > the case that the more expressive the model is, the harder it is to
> > interpret and to explain its decisions? Since you claim that your work
> > paves the way between XAI and formal verification, I think that the
> > trade-off I'm mentioning should be investigated.
> >
> > > This is why we believe our paper is interesting:it proposes a new
> > > and second approach to convert DNNs into SAT formulas.
> >
> > More precisely, it proposes a novel convolutional architecture that can be
> > converted into SAT formulas, not a general conversion method for DNNs.
> >
> > > Only BNNs can be transformed into SAT at this time. In a way what
> > > you say is right: {-1,0,1} are particular reals but our method
> > > generalizes to any kind of real weight. We refer the reviewer to
> > > works [2] and [3] for the conversion to SAT.
> >
> > You are right, when writing that comment I missed "real-valued" DCNN.
> > Anyhow, wouldn't it be possible (although surely not practical) to encode
> > a (real-valued) ReLU network as a SMT formula and then consider its boolean
> > abstraction + an eager encoding of the algebraic theory? If so, I
> > would mention that your architecture enables a practical conversion.

---

> > > ### Author Response · Authors · 2021-11-22
> > > **Answer**
> > >
> > > We would like to thank the reviewer for his detailed comments on our submission.
> > >
> > > We will take these comments into account in the next version. In particular, we will remove the strong statements about the interpretability:
> > >
> > > "easily interpreted",
> > > "We believe our new architecture paves the way between eXplainability AI (XAI) and formal verification."
> > > "highly interpretable"

---

> > > > ### Comment · Reviewer_UdYz · 2021-12-02
> > > > **Re: answer**
> > > >
> > > > > We will take these comments into account in the next version. In particular, we will remove the strong statements about the
> > > > > interpretability:
> > > > >"easily interpreted", "We believe our new architecture paves the way between eXplainability AI (XAI) and formal verification." >"highly interpretable"
> > > >
> > > > These claims are still present in the current version, which doesn't address the points raised in my initial review.
> > > > I cannot change my initial score. This work has potential but its presentation is currently burdened by unsubstantiated XAI claims that can confuse the reader.

---

> > > > > ### Author Response · Authors · 2021-12-02
> > > > > **Answer**
> > > > >
> > > > > Thank you for your reply, we did not update the version as we will resubmit our paper to an another conference, taking into account your comments of course.

---

### Official Review · Reviewer_wThs · 2021-11-01

**Correctness:** 3
**Technical Novelty And Significance:** 2
**Empirical Novelty And Significance:** 2
**Recommendation:** 3
**Confidence:** 4

**Main Review:**

**Strength**
This work has proposed a method to improve the scalability of SAT solvers to deep networks to aid robustness verification.

**Weakness**
* The models used in this paper are for very small-sized neural networks trained on simple datasets like MNIST and CIFAR 10. This approach is unlikely to ever scale to large models with millions of parameters trained on datasets like Imagenet. The main reason being *grouping parameters* makes a trade-off between the generalizability of the network and its scalability to SAT solvers.
* This paper uses *grouping parameters* to keep things scalable. This compromises the overall accuracy of the model but prevents the number of clauses from exploding. Even with this simplification, this method performs marginally better than the real value-based complete verification method for low noise is unable to perform better in high noise cases such as *MNIST epsilon_{test} = 0.3* and *CIFAR 10 epsion_{test} = 8/255*.
* The statement *The only scalable method for global exact interpretability was proposed in (Granmo et al., 2019)* in sec. 2  is somewhat inaccurate.  [1] is a recent work that addresses the issue of scalability of MIP solvers to Deep Networks in the context of Model Explanation. This work essentially simplifies the formula to be encoded in the MIP solver by using gradient-based methods. The MIP solver then generates explanations by solving for the simplified formulation. This simplified formulation is what allows this method to scale to very deep networks typically feat. millions of parameters.

**citations**

[1] Subham Sekhar Sahoo, Subhashini Venugopalan, Li Li, Rishabh Singh, and Patrick Riley. Scaling Symbolic Methods
using Gradients for Neural Model Explanation. International Conference on Learning Representations, 2021.
https://arxiv.org/abs/2006.16322

**Summary Of The Paper:**

This paper proposes a new kind of SAT-encodable Neural Network which has real-valued weights and binary activations. Since the activations are binary, this work constructs a truth table to map the inputs of a CNN block to its outputs. This helps to deduce the output of the CNN block as a boolean function of the input variables while allowing intermediate values to be real. This formula also gives a set of masks that identify the conditions for which a given neuron is active. Tuning this set of masks post training helps increase the generalisability of the network - sec 4.1. Furthermore, this causes the encoded formula to be relatively simpler than the other existing methods for verifying BNNs and thus helps in keeping things scalable for the sat solver. The authors have conducted experiments to show that this method achieves a better performance in terms of verifiable accuracy and runtime than the existing methods (Table 1).

**Summary Of The Review:**

Working with SAT solvers and Neural Networks is a very challenging task. [1] has already set the bar high and has paved the way for scaling NP-Hard solvers like MIP solvers to very deep neural networks. Unless the authors can achieve a similar feat where their approach scales to much deeper networks (ex. Inception model), I'm leaning towards rejecting this work because I see this work as incremental.

**citations**

[1] Subham Sekhar Sahoo, Subhashini Venugopalan, Li Li, Rishabh Singh, and Patrick Riley. Scaling Symbolic Methods
using Gradients for Neural Model Explanation. International Conference on Learning Representations, 2021.
https://arxiv.org/abs/2006.16322

---

> ### Author Response · Authors · 2021-11-15
> **We would like to thank the reviewer for his detailed comments on our submission.  We will try to answer every point you highlight step by step.**
>
> > "This approach is [...] Imagenet"
>
> You are right about the trade-off between generalisation and scalability. However, we disagree that our model is unlikely to scale. One of our future work is to investigate this.
>
> Furthermore, while we fully understand the reviewer's concern about scalability, **we emphasize that our paper is not about scaling a model**. It is about the formal and complete verification of the robustness of a DNN. Today, the best verifiable accuracy on the smallest image dataset MNIST with the smallest noise investigated (0.1 - imperceptible to the eye) is** 95.62% and it takes 3.52s to infer.  We propose a comparable verifiable accuracy, 10 times faster**. For CIFAR10 high noise (8/255, imperceptible to the eye) state-of-the-art is **22.55% (baseline=10%) verifiable accuracy in 0.1781s with a custom made SAT solver. We propose 23.08% (+0.53%) in 0.3887s, with a general SAT solver.**
>
> **In this context, we don't see scaling up as the  priority : because even on small datasets, with small models, the problem of exact verification of DNNs is open.**
>
> Finally, we note that your remark can be applied on all previous works on the domain of formal verification of DNN with SAT solvers [2], [3], [4]. For example, CIFAR10 was only introduced in [2, NeurIPS (2020)].
>
>
>
> > "The main reason [...] SAT solvers."
>
> > "This paper [...] things scalable."
>
>
> You are perfectly right.We will add it to the final version.
>
>  > "Even with [...] for low noise"
>
> We partially disagree with the reviewer: for low noise MNIST, our verifiable accuracy is 94.26% and verification time is 0.3724s vs  the best real value-based complete verification method, the verifiable accuracy is 95.62%  in 3.52s. **So we do-1.4% less good but 10 times faster.**
>
> Besides, for low noise CIFAR10, our verifiable accuracy is 33.04%  in 0.7782s, with a timeout of 0%. vs the best real value-based complete verification method, the verifiable accuracy is 45.93%  in 66.08s, with a timeout of 1.86%. **So we do -12.9% less good (which is a lot we confess) but 100 times faster. And we don't have any timeouts which allows our method to be complete**
>
>
>   > " [...] is unable  [...] epsion_{test} = 8/255. "
>
> On this point we totally disagree with the reviewer: for high noise MNIST, our verifiable accuracy is 80.36% and verification time is 0.5722s, with a timeout of 0%. vs the best real value-based complete verification method, the verifiable accuracy is 80.68%%  in 7.12 s (please note it is not the same paper as above for MNIST), with a timeout of 2.47%. **Thus, for a similar accuracy, we are 12 times faster and without any timeout.**
>
> For high noise CIFAR10, our verifiable accuracy is 23.08% and verification time is 0.3887 vs for the best real value-based complete verification method, the verifiable accuracy is 20.27% in 60.67s, with a timeout of 2.47%. **Thus, in this case we are much more accurate, much faster and without any timeout.**
>
>
> **Our paper is a trade-off between a real value-based complete verification method (we are better in time and interpretability) and the previous SAT-based complete verification method (we are better in verifiable accuracy and interpretability).**
>
>  > "The statement [...] 2 is somewhat inaccurate."
>
> We would like to thank the reviewer for pointing this reference, which we will add.
>
> However, we disagree on 2 points:
>
>  1- **First, this paper does not do any formal verification. It is an only an interpretation paper**, it never measures the existence or not of adversarial attack for a given model, image and noise level. Just because they use a SMT does not mean that their work necessarily falls into the field of formal verification. As written earlier, to the best of our knowledge, all the best formal verification works are cited and compared in our work.  We will add [3] at the request of the reviewer 2, but as our reply indicates, we far surpass this baseline.
>
> 2 - Secondly, while it seems possible that their work on global formal interpretation could be extended to the field of formal verification, this will be **non-exact** verification (whereas our work falls into the **exact** domain). **Indeed, before the exact verification by SMT, the selection of the pixels to be verified is done by a non-exact method (IG based).**
>
>
>
> We understand that this point supports the reviewer's remark 1 on the scalability issue of our paper. Thus, we hope that, if the reviewer agrees with our rebuttal, he/she will acknowledge that, to date, no formal exact verification paper has been proposed.
>
>
>
> [1] Approximating CNNs with Bag-of-local-Features models works surprisingly well on ImageNet, Brendel et al.
>
> [2] Efficient exact verification of binarized neural networks, Jia & Rinard (2020) (NeurIPS 2020)
>
> [3] In search for a sat-friendly binarized neural network architecture, Narodytska et al. (ICLR 2020)
>
> [4] Verifying properties of binarized deep neural networks, Narodytska et al. (AAAI2018)

---

### Official Review · Reviewer_garj · 2021-11-02

**Correctness:** 3
**Technical Novelty And Significance:** 3
**Empirical Novelty And Significance:** 2
**Recommendation:** 5
**Confidence:** 4

**Main Review:**

Strengths:
- the idea of using a truth table to distill (or approximate) real-valued feature vector is novel and very interesting
- TT-DCNN provides certain interpretability and supports manual post-processing
- the robustness of TT-DCNN can be verified using SAT solvers

Weaknesses and actionable feedback:
- the writing is a bit hard to follow, which is mainly due to frequently referring to the appendices that seem necessary to understand the main content of this paper (e.g., Sec 3.4, V(f,i,j) is not easy to get without carefully looking back and forth). Heavily using appendices in such a way adds extra burden for reviewers and gains unfair advantages.
- missing important baselines, for instance, the authors are clearly aware of that Narodytska et al. 2019b work is highly related, given it has been cited many times; however, it is a bit surprising to see the evaluation does not use it as a baseline.
- there are lots of conceptual claims, however, which are not backed up by experimental results. Sometimes the result indicates the opposite.  For instance, "drastic reduction in the size of SAT formulas renders our model truly amenable to formal verification". This is not true, since the evaluation shows that the baseline Jia & Rinard (2020) is actually much faster (e.g., 3~30 times faster). For another instance, the authors emphasize "highly interpretable" lots of times, however, simply filtering out sparse binary masks seems not sufficient to support that claim, at least some case studies (e.g., what highly interpretable features are actually learned) need to be presented in order to support that claim.
- TT-DCNN seems to be a new architecture, and some discussion regarding training would be necessary.


**Summary Of The Paper:**

This paper proposes Truth Table Deep Convolutional Neural Networks (TT-DCNNs), which _distills_ real-valued feature vectors (i.e., small convolutional layers) into truth tables (also called masks), which are represented in boolean formulas using the disjunctive normal form (DNF). These masks indicate certain interpretability, e.g., the set of variables actually contributed to the truth value. Some manual intervention or post processing like removing overfitting masks could help improve model accuracy. The experimental evaluation shows that model accucy on MNIST and CIFAR10 dataset is comparable with or slightly better than other state-of-the-art approaches.

**Summary Of The Review:**

I like the idea presented in this paper and hope it could be backed up in a more systematic and scientific way. Writing issues should be easy to address; it is just the current evaluation is a bit unsatisfactory (i.e., missing a highly relevant and important baseline), thus not very convincing. So I slightly prefer not to accept this paper, at least in its current state. I would be happy to adjust my assessment, if the authors could address my concerns about experimental evaluation during rebuttal.

---

> ### Author Response · Authors · 2021-11-15
> **We would like to thank the reviewer for his detailed comments on our submission.  We will try to answer every point you highlight step by step.**
>
> > "Heavily using [...] unfair advantages."
>
> The editorial comments will be taken into account in the next version.
>
> > "missing important  [...]  a baseline. "
>
> We would like to clarify that **it is not that we have forgotten this comparison, but it is that the results presented in [1 - Narodytska et al. 2019b] are far below those proposed by [2 - Jia & Rinard (2020) (NeurIPS 2020)].** Thus, we preferred to compare our work to the current state of the art.
>
> In [1], the authors only authorize at **maximum 20 pixels to switch**, no more (cf page 8, section 6 EXPERIMENTS, paragraph untargeted attack), whereas in our case, for high noise MNIST, we have on average 40 pixels that switch (between 18 to 85 pixels).
>
> Checking Figures 1 and 2 in [1], one can observe that even with a 10 pixels attack (which is the average of pixels that switch for 0.1 noise), it takes 5 seconds to verify their SAT equation and the success ratio of the attack is approximately 80% (results done on only 100 images) which means a verifiable accuracy of 20%.
>
> In our case, if we authorize only 10 pixels to switch, we get a verifiable accuracy of 95.7% (so +65% compared to [1]) with a verification time of 0.159s (so x50 faster compared to [1]).
>
> This comparison  is summarized below and we will include it in our paper.
>
>
> |        |      |            |                               |   Accuracy  |         |   | Mean time |   | Timeout |   | #cls/#vars |
> |:------:|:----:|:----------:|:-----------------------------:|:-----------:|:-------:|:-:|:---------:|:-:|:-------:|:-:|:----------:|
> |        |      |            |                               | Verifiable  | Natural |   |           |   |         |   |            |
> |  MNIST |  noise:10 pixels |  SAT-based |         TT-DCNN (Ours)        |    95.7 %   | 96.79 % |   |   0.159   |   |    0    |   |   1K/0.4K  |
> |        |      |            | [2 - Narodytska et al. 2019b] |     20%     |  96.0 % |   |     5     |   |    0    |   |   8K/20K   |
>
>
>
>   > "there are lots [...]  3~30 times faster). "
>
>  In the case you mention, we may not have been precise enough because in the case of [2, Jia & Rinard (2020)], **the SAT solver is specific to their problem, while in our case we use a  general one** (cf solver MiniCard, specified in first paragraph of section 4.3 of our paper).
>
> This point is important, as here are the results if one uses the same general SAT solvers (MiniSat): Table 8 in [2] presented for conv-large has a __solving time__ of 0.242s (for MNIST high noise) vs our __solving time__ which is 0,00079s (for MNIST high noise). **Therefore, we are about 300 fasterif we use the same generic SAT solver.**
>
> Please note that the Solve time (presented here) is different from mean time (presented in the paper): in the paper we count the build time (the one needed to construct the SAT equation) + the solve time (time needed to solve the equation). Again, we took the same numbers as the paper to be fair: their work proposes a new specificly crafted SAT solver, we prefer to use a general one.
>
> We also want to highlight that this order of magnitude is not surprising, as our CNF equation is 48 times smaller in terms of literals and 5 times smaller in terms of clauses on average compared to [2].
>
>
>  > "For another  [...]   to support that claim."
>
> We agree with the reviewer and thus we will replace “highly interpretable” by “more interpretable than BNN conversion”. We would like to investigate in further work what these features are. However, for this paper, we already have too much content.
>
> We would like to also emphasize that our work is not a paper on interpretability - it is a paper on designing a new method to convert a DNN into SAT equations and what are the results on formal and complete robustness verification.
>
>  > " TT-CNN seems to be [...]  be necessary. "
>
> **You are perfectly right, and this is exactly what we would like to investigate in future work.** However, again we believe that for this paper we already have too much content.
>
> Furthermore, our original goal was not to be scalable (at least not more than the state-of-the-art for verification), it was to be a compromise between the real valued approach and the BNN approach for exact formal verification of DNN and to start a small bridge between interpretability and formal verification. All this by presenting a new approach.
>
> **We would like to point out that there was no ambiguity or dissimulation, we have worked with exactly the same training set up as [2], see Appendix A6 in our paper, paragraph Training method, first sentence.**
>
>
> [1] In search for a sat-friendly binarized neural network architecture, Narodytska et al. (ICLR 2020)
>
> [2] Efficient exact verification of binarized neural networks, Jia & Rinard (2020) (NeurIPS 2020)

---

### Official Review · Reviewer_RhEH · 2021-11-03

**Correctness:** 2
**Technical Novelty And Significance:** 2
**Empirical Novelty And Significance:** 2
**Recommendation:** 3
**Confidence:** 4

**Main Review:**

In my opinion, the paper's premises are not correct. As far as I can
understand, BNNs were developed to target issues like power
consumption and ease of circuit-level integration, and not the
difficulties with propositional encodings or the sizes of such
encodings, or even issues interpretability, of the models or their
encodings.

Furthermore, I cannot understand why interpretability of a SAT
encoding is a concern. The SAT encoding is simply a representation
that is used to reason about the original machine learning model.
SAT encodings are important because they enable efficient reasoning in
some settings, but not because of issues with interpretability.
Furthermore, any information resulting from reasoning about the
original machine learning model on such a representation can be
related with the original model.

To obtain a SAT encoding the paper makes several critical decisions.
One is to consider features to take boolean values. Another one is to
assume a very restricted representation, one where at most nine
features can be considered. From what I can understand, raising
significantly the number of features used in the representation would
make the encoding impossibly large.

In my opinion the paper is not well written. For example, 2D-CNNs are
briefly described in a paragraph containing no references. The
explanation is not easy to follow. This applies to section 2, but also
to section 3. For example, I was unable to understand how one goes
from equation (1) to an intermediate equation claiming that one can
generate both CNF and DNF representations. I was unable to make sense
of the subsequent example, and I have some experience with logical
encodings.

Also, the integration of different blocks is unclear. I can understand
how this might be done in the case of an intermediate CNF
representation. However, I was unable to understand how this might be
done for a DNF intermediate representation. If this was easy to do,
then satisfiability for CNNs should be easy to decide, and that cannot
be the case. I could be missing something here, but the writeup does
not help.


**Summary Of The Paper:**

The paper proposes a new machine learning model TT-DCNN as an
alternative to Binarized Neural Networks (BNNs). The paper claims that
the new machine learning model simplifies and improves
interpretability of the SAT encodings.


**Summary Of The Review:**

The paper proposes to learn a new machine learning model, one that
simplifies the propositional encoding, and subsequent reasoning with a
SAT solver. The work could be of importance, but the current writeup
does not help to convey the key ideas. In its present form, I cannot
recommend acceptance of this paper.

---

> ### Author Response · Authors · 2021-11-15
> **We would like to thank the reviewer for his detailed comments on our submission.  We will try to answer every point you highlight step by step.**
>
>
>
> > "As far as [...] their encodings."
>
> You are right on your point of BNN. However, **we never claim that BNN were designed for SAT encoding**, as you suggested. And we would not be able to say it: the original paper [1] was published in 2016 and the first conversion of BNN into SAT was published in 2018 [2].
>
>
>  > "Furthermore, [...] a concern."
>
> We disagree that this is not a concern: as  the understanding of SAT conversion is a necessary brick in order to make a bridge between formal verification and XAI.
>
> To transform an inequality into SAT formula, you have to apply the cardinality constraint, which is a very opaque operation. An example of the conversion is given in Table 2 of our article: for encoding x_1 - 2 x_2 + 3 x_3 <= 3, the formula CNF is (l_4) \land (\overline{l_1} \lor l_2 \lor \overline{l_5}) \land (l_5 \lor \overline{l_3} \lor \overline{l_6}) \land (l_6) . Whereas, we can make sense of the inequality x_1 - 2 x_2 + 3 x_3 <= 3: the filter of size 2x2: [[1,-2][3,0]] with a bias of -3, then a ReLu function gives the inequality above, if you consider the input [[x1, x2][x3,x4]]. But then, when you turn it into a CNF formula, we can't make sense of either the literals or the clauses. What does l_4 represent in relation to inputs and filter? What about the (\overline{l_1} \lor l_2 \lor \overline{l_5})  clause? This is very hard to say. And therefore, when you put this equation into the sat solver, oneis able to verify the DNN but not to make sense of the SAT equation.
>
> Our paper tries to solve this current limit of the formal verification field.
>
>
> > "From what I can understand, [...] impossibly large."
>
> You point out in fact a limit of our model. However, with n <= 9 only, we manage to be competitive with the BNN natural accuracy, the one desired. Therefore, for the moment, it is not a limitation. Also, Table 6 gives you better insight of the importance of n on the natural accuracy. Finally, we can go up to 48 without problem.
>
>
>  > "For example, [...]  references."
>
> The editorial comments will be taken into account in the next version.
>
> > "For example, [...]  logical encodings."
>
> We did not want to spend time on the transition from the truth table to CNF/DNF equations as it seems to us quite standard.
> We refer the reviewer to the following course for example https://www.csd.uwo.ca/~mmorenom/cs2209_moreno/slide/lec8-9-NF.pdf (slides 9 to 13), which explains the passage in details, and we will add an appropriate reference.
>
>
> > "Also, [...]  help."
>
> Sorry, we are not able to understand this paragraph.
>
> Of which *"integration"* do you refer to?
>
> *"I can understand [...] intermediate CNF representation. However, [...] representation."*
>
> If the reviewer understands the intermediate CNF representation, we don't understand how he doesn't understand the DNF ones. We want to recall that one can transform any CNF into a DNF (you just need to develop). Example: A \land (B \lor C) = (A \land B) \lor (A \land C).
>
> We are not sure what is meant by *"satisfiability for CNNs should be easy to decide"*? There is no satisfiability for CNNs. We are not sure what is meant by *"decide"*?
>
> ## Remarks:
>
> Finally, **we want to point out that the reviewer did not mention any of our results, despite the fact that we outperform the NeurIPS 2020 paper [1] on verified accuracy**: +2.8% for MNIST high noise, +0.5 CIFAR10 high noise (for the same computation time despite the fact that we use a general solver in our case versus a specific one in their case).
>
>
> ## Conclusion:
>
> In general, we fully understand the reviewer's grievances about our article: **some editorial points can be improved**. However, we very regret that this work, which focuses on improving the trade-off between verifiable accuracy and verification time while bringing an interpretative dimension to the whole, is given a straight reject grade.  Moreover, our work is the only one, to the best of our knowledge, that proposes an improvement over the full SAT verification since NeurIPS 2020 [1], all through a new approach.
>
>
> [1] Binarized Neural Networks, Hubara et al.
>
> [2] Verifying properties of binarized deep neural networks, Narodytska et al.

---

> > ### Comment · Reviewer_RhEH · 2021-12-01
> > **Final comment**
> >
> > I re-read my review and I read the rebuttal. The review contains a few minor typos, but all in all, I stand by what I wrote. The paper should be rewritten, its main claims should be sharpened. Overall the writeup should be improved, and then resubmitted. In its current form, I cannot recommend acceptance.

---

### Decision · Program_Chairs · 2022-01-20

**Decision:**

Reject

**Comment:**

I think there is good research behind this paper, but the presentation issues make it difficult to argue for acceptance.

On the positive side, the paper has made a clear advance in terms of the ability to do full SAT-based verification of neural networks. However, there are also important issues with the paper that prevent it from being accepted:
* The paper argues for the value of the new approach for *both* verifiability and interpretability, where interpretability is measured in terms of the ability to make targeted adjustments to the network to change its behavior. These are very different goals, but they are conflated in different parts of the paper, leading to confusion, for example, from reviewer RhEH.
* The paper only compares against SAT/SMT-based verification, but completely ignores other approaches to verification that are arguably more effective for many problems. In particular, there is an emerging literature on Abstract Interpretation-based verification that is significantly more scalable than SAT-based verification and which this paper ignores.
* The paper's claims sometimes get ahead of the presented evidence, as pointed out by reviewer garj.

So overall, I think this paper needs another iteration before it is ready for acceptance.